# Gaussian-Based Pooling for Convolutional Neural Networks

**Takumi Kobayashi**
National Institute of Advanced Industrial Science and Technology (AIST)
1-1-1 Umezono, Tsukuba, Japan
`takumi.kobayashi@aist.go.jp`

## Abstract

Convolutional neural networks (CNNs) contain local pooling to effectively down-size feature maps for increasing computation efficiency as well as robustness to input variations. The local pooling methods are generally formulated in a form of convex combination of local neuron activations for retaining the characteristics of an input feature map in a manner similar to image downscaling. In this paper, to improve performance of CNNs, we propose a novel local pooling method based on the Gaussian-based probabilistic model over local neuron activations for flexibly pooling (extracting) features, in contrast to the previous model restricting the output within the convex hull of local neurons. In the proposed method, the local neuron activations are aggregated into the statistics of mean and standard deviation in a Gaussian distribution, and then on the basis of those statistics, we construct the probabilistic model suitable for the pooling in accordance with the knowledge about local pooling in CNNs. Through the probabilistic model equipped with trainable parameters, the proposed method naturally integrates two schemes of adaptively training the pooling form based on input feature maps and stochastically performing the pooling throughout the end-to-end learning. The experimental results on image classification demonstrate that the proposed method favorably improves performance of various CNNs in comparison with the other pooling methods. The code is available at `https://github.com/tk1980/GaussianPooling`.

## 1 Introduction

In recent years, convolutional neural networks (CNNs) are applied to various visual recognition tasks with great success [7, 8, 14]. Much research effort has been made in improving the CNN architecture [7, 8] as well as the building blocks of CNNs [6, 11, 24, 29]. Local pooling is also a key component of CNNs to downsize feature maps for increasing computational efficiency and robustness to input variations.

From a biological viewpoint, the local pooling originates from the neuroscientific study on visual cortex [10]. While some works biologically suggest the importance of *max*-pooling [20, 21, 23], *average*-pooling also works for some CNNs in practice, and thus we can say that the optimal pooling form is dependent on the type of CNN, dataset and task. To improve performance of CNNs, those simple pooling methods are sophisticated by introducing some prior models related to pooling. Based on the pooling functionality which is akin to image downsizing, some image processing techniques are applied to the pooling operation such as Wavelet [18] for Wavelet pooling [28] and image downscaling method [27] for detailed-preserving pooling (DPP) [22]. On the other hand, by focusing on the pooling formulation, the mixed-pooling and gated-pooling are proposed in [15, 32] by linearly combining the average- and max-pooling. Recently, in [1], the local pooling is formulated based on the maximum entropy principle. Those methods [1, 15, 22] also provide trainable pooling forms equipped with pooling parameters which are optimized throughout the end-to-end learning;

especially, the scheme of global feature guided pooling (GFGP) [1] harnesses the input feature map for adaptively estimating the pooling parameters. Besides those deterministic methods, the stochastic pooling is also proposed in [33] to introduce randomness into the local pooling process with similar motivation to DropOut [24] toward improving generalization performance. Such a stochastic scheme can be applied to the mixed-pooling by stochastically mixing the average- and max-pooling with a random weight [15, 32].

The above-mentioned pooling methods are generally described by a *convex-hull* model which produces the output activation as a convex combination of the input neuron activations (Section 2.1). This model is basically derived from image downscaling to reduce the spatial image size while approximating an input image to maintain image content or quality [22]. However, the convex-hull model is not crucial for extracting features in CNNs, and practically speaking, high-performance recognition does not strictly demand to well approximate the input feature map at the process of local pooling. Therefore, the local pooling operation would be formulated more flexibly for improving performance of CNNs.

In this paper, we propose a novel local pooling method by considering the probabilistic model over the local neuron activations, beyond the sample-wise representation in the previous convex-hull formulation. In the proposed method, to summarize the local neuron activations, we first assume a Gaussian distribution for the local activations and thereby aggregate the activations into the two simple statistics of mean and standard deviation. This is just a process to fit the Gaussian model to the input neuron activations, and then we modify the Gaussian model into the probabilistic model suitable for pooling such that the pooling output can be described more flexibly based on the local statistics with trainable parameters. In accordance with the knowledge about local pooling in CNNs [1, 15, 22], we propose the model of the inverse softplus-Gaussian distribution to formulate the trainable local pooling. Thus, the proposed pooling method naturally unifies the stochastic training in local pooling [33] and the adaptive parameter estimation [1] through the parameterized probabilistic model; these two schemes are complementary since the stochastic training boosts the effectiveness of the trainable pooling model which renders discriminative power to CNNs with a slight risk of over-fitting.

## 2 Gaussian-based pooling

We first briefly review the basic pooling formulation on which most of the previous methods [1, 15, 22, 32, 33] are built. Then, the proposed pooling methods are formulated by means of probabilistic models to represent the output (pooled) activation more flexibly.

### 2.1 Convex-hull model for pooling

Most of the local pooling methods, including average- and max-pooling, can be reduced to a linear convex combination of local neuron activations, which is regarded as a natural model from the viewpoint of minimizing the information loss caused by downsizing feature maps as in image downscaling. The convex-combination model is formulated as follows. The local pooling operates on the $c$-th channel map of an input feature tensor $\boldsymbol{X} \in \mathbb{R}^{H \times W \times C}$ (Fig. 1a) by

$$Y_{\boldsymbol{q}}^c = \sum_{\boldsymbol{p} \in \mathcal{R}_{\boldsymbol{q}}} w_{\boldsymbol{p}}^c X_{\boldsymbol{p}}^c, \quad s.t. \sum_{\boldsymbol{p} \in \mathcal{R}_{\boldsymbol{q}}} w_{\boldsymbol{p}}^c = 1, \ w_{\boldsymbol{p}}^c \geq 0, \forall \boldsymbol{p} \in \mathcal{R}_{\boldsymbol{q}}, \tag{1}$$

where $\boldsymbol{p}$ and $\boldsymbol{q}$ indicate the 2-D positions on the input and output feature map, respectively, and the receptive field of the output $Y_{\boldsymbol{q}}^c$ is denoted by $\mathcal{R}_{\boldsymbol{q}}$; these notations are also depicted in Fig. 1a. The local neuron activations $\{X_{\boldsymbol{p}}^c\}_{\boldsymbol{p} \in \mathcal{R}_{\boldsymbol{q}}}$ are aggregated into the output $Y_{\boldsymbol{q}}^c$ by using the *convex* weights $\{w_{\boldsymbol{p}}^c\}_{\boldsymbol{p} \in \mathcal{R}_{\boldsymbol{q}}}$; in other words, $Y_{\boldsymbol{q}}^c$ is restricted to the convex hull of $\{X_{\boldsymbol{p}}^c\}_{\boldsymbol{p} \in \mathcal{R}_{\boldsymbol{q}}}$ (Fig. 1b). In this model, the convex weight characterizes pooling functionality. For instance, average-pooling employs $w_{\boldsymbol{p}}^c = \frac{1}{|\mathcal{R}_{\boldsymbol{q}}|}$ while max-pooling only activates the single weight of the most prominent neuron, and those two types of weights can be mixed [15, 32]. The convex weights can be defined in a sophisticated way such as by introducing the image processing technique [22] and the maximum entropy principle [1] to provide a trainable pooling form. The Gaussian model is also introduced to construct the convex weights in [25] similarly to softmax. In the stochastic pooling [33], the multinomial probabilistic model is applied to the weights by setting $w_{\boldsymbol{p}}^c = \frac{X_{\boldsymbol{p}}^c}{\sum_{\boldsymbol{p}'} X_{\boldsymbol{p}'}^c}$ and the method

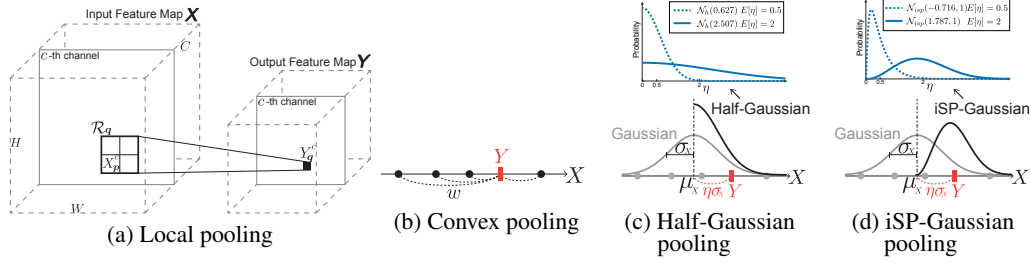

(a) Local pooling  (b) Convex pooling  (c) Half-Gaussian pooling  (d) iSP-Gaussian pooling

Figure 1: Local pooling operation in CNN. The pooling downsizes an input feature map through locally aggregating activations (a). The previous pooling methods aggregate input neuron activations $X$ with convex weights $w$, thus restricting the output $Y$ to the convex hull of $X$ (b). On the other hand, the proposed Gaussian-based pooling outputs $Y$ according to the half-Gaussian distribution (c) or inverse softplus (iSP)-Gaussian (d) which utilize the two statistics of mean $\mu_X$ and standard deviation $\sigma_X$ of the input local activations $X$.

stochastically outputs $Y_q^c = X_p^c$ according to the probability $w_p^c$ in the training. As to the stochastic scheme in local pooling, $S^3$ pooling [34] embeds randomness into the selection of the receptive field $\mathcal{R}_q$ for the output $Y_q^c$.

## 2.2 Half-Gaussian pooling

The form of convex combination in Eq. 1 is effective for *image* downscaling while keeping image quality, but is not necessarily a crucial factor for pooling to downsize *feature* maps in CNNs; for better recognition by CNNs, we can freely produce $Y_q^c$ beyond the convex hull of inputs $\{X_p^c\}_{p \in \mathcal{R}_q}$. Thus, we formulate *Gaussian-based Pooling* to describe the output by means of probabilistic models beyond the sample-wise representation in Eq. 1. We hereafter omit the superscript $c$ (channel) and subscript $q$ (output position) for simplicity; Table 1 summarizes the detailed forms of the methods.

First, the local neuron activations $\{X_p\}_{p \in \mathcal{R}}$ are modeled by a Gaussian distribution with the mean $\mu_X$ and standard deviation $\sigma_X$;

$$\tilde{X} \sim \mathcal{N}(\mu_X, \sigma_X) \Leftrightarrow \tilde{X} = \mu_X + \epsilon \sigma_X, \tag{2}$$

$$\text{where } \mu_X = \frac{1}{|\mathcal{R}|} \sum_{p \in \mathcal{R}} X_p, \ \sigma_X^2 = \frac{1}{|\mathcal{R}|} \sum_{p \in \mathcal{R}} (X_p - \mu_X)^2, \ \epsilon \sim \mathcal{N}(0,1), \ \epsilon \in (-\infty, +\infty). \tag{3}$$

This, however, provides just a model to probabilistically *reproduce* the local neuron activations. We thus modify the Gaussian model in Eq. 2 into the ones suitable for local pooling in CNNs. As empirically shown in [1] and suggested in [15, 22, 32], the pooling whose functionality is biased toward *min* below *average* is less effective in providing discriminative feature representation since it suppresses neuron activations, degrading performance. Based on the knowledge about local pooling, we can modify Eq. 2 into, by prohibiting the output from falling below the mean $\mu_X$,

$$Y = \mu_X + |\epsilon|\sigma_X, \ \epsilon \sim \mathcal{N}(0,1) \Leftrightarrow Y = \mu_X + \eta\sigma_X, \ \eta \sim \mathcal{N}_h(1), \ \eta \in [0, +\infty), \tag{4}$$

where the *half-Gaussian* distribution $\mathcal{N}_h(\sigma_0)$ [19] (Fig. 1c) with $\sigma_0 = 1$ is naturally introduced as a prior probabilistic model; note that $\mathrm{E}[\eta] = \sigma_0 \frac{\sqrt{2}}{\sqrt{\pi}}$ and $\mathrm{Var}[\eta] = \sigma_0^2(1 - \frac{2}{\pi})$ for $\eta \sim \mathcal{N}_h(\sigma_0)$. Thereby, the fixed *half-Gaussian pooling* is formulated in Eq. 4 to stochastically produce $Y$ without using any pooling parameter, and at an inference phase the pooling works in a deterministic way by utilizing the mean of $\mathcal{N}_h(1)$ as $Y = \mu_X + \frac{\sqrt{2}}{\sqrt{\pi}}\sigma_X$.

**Parametric pooling**  We then extend the fixed half-Gaussian pooling in Eq. 4 by introducing a variable parameter $\sigma_0$, the standard deviation of the half-Gaussian, to flexibly describe the output;

$$Y = \mu_X + \eta\sigma_X, \ \text{where } \eta \sim \mathcal{N}_h(\sigma_0), \ \eta \in [0, +\infty), \ \sigma_0 = \texttt{softplus} \circ \texttt{f}(\boldsymbol{X}) \tag{5}$$

$$\Leftrightarrow Y = \mu_X + |\epsilon|\sigma_0\sigma_X, \ \text{where } \epsilon \sim \mathcal{N}(0,1), \ \epsilon \in (-\infty, +\infty), \ \sigma_0 = \texttt{softplus} \circ \texttt{f}(\boldsymbol{X}), \tag{6}$$

where the parameter $\sigma_0$ is estimated from the input feature map $\boldsymbol{X}$ by the GFGP method [1];

$$\sigma_0 = \texttt{softplus} \circ \texttt{f}(\boldsymbol{X}) = \texttt{softplus}(b + \boldsymbol{v}^\top \texttt{ReLU}(\boldsymbol{a} + \boldsymbol{U}^\top \texttt{GAP}(\boldsymbol{X}))), \tag{7}$$

where $\text{GAP}(\boldsymbol{X}) = \frac{1}{HW} \sum_{\boldsymbol{p}=(1,1)}^{(W,H)} \boldsymbol{x_p} \in \mathbb{R}^C$ is the global average pooling (GAP) [17], $\{\boldsymbol{U}, \boldsymbol{a}\} \in \{\mathbb{R}^{C \times \frac{C}{2}}, \mathbb{R}^{\frac{C}{2}}\}$ and $\{\boldsymbol{v}, b\} \in \{\mathbb{R}^{\frac{C}{2}}, \mathbb{R}\}$ are the parameters of the two-layered MLP in the GFGP [1], and the softplus function $\text{softplus}(x) = \log\{1 + \exp(x)\}$ is applied to ensure the non-negative $\sigma_0$. The deterministic pooling for inference is accordingly given by

$$Y = \mu_x + \frac{\sqrt{2}}{\sqrt{\pi}} \sigma_0 \sigma_x, \quad \text{where } \sigma_0 = \text{softplus} \circ \text{f}(\boldsymbol{X}). \tag{8}$$

The flexible half-Gaussian pooling (Fig. 1c) in Eq. 6 allows the output to be far from the mean $\mu_x$ possibly beyond $\max_{\boldsymbol{p} \in \mathcal{R}}(X_{\boldsymbol{p}})$, and the deviation from the mean is controlled by the parameter $\sigma_0$ which is estimated in Eq. 7 exploiting the global features $\boldsymbol{X}$; the effectiveness of estimating local pooling parameters from global features is shown in [1]. It is noteworthy that in the proposed method, the parametric half-Gaussian model naturally incorporates the parameter estimation by GFGP [1] with the stochastic pooling scheme.

## 2.3   Inverse softplus-Gaussian pooling

Though the half-Gaussian model is derived from the Gaussian distribution of neuron activations as described in Section 2.2, the model is slightly less flexible in that the single parameter $\sigma_0$ tightly couples the mean and variance of the half-Gaussian distribution by $\text{Var}[\eta] = \sigma_0^2(1 - \frac{2}{\pi}) = \text{E}[\eta]^2(\frac{\pi}{2} - 1)$ for $\eta \sim \mathcal{N}_h(\sigma_0)$; it inevitably enlarges the variance for the larger mean (Fig. 1c). To endow the pooling model with more flexibility, we propose *inverse softplus*-Gaussian (iSP-Gaussian) distribution[1] in

$$\eta \sim \mathcal{N}_{isp}(\mu_0, \sigma_0) \Leftrightarrow \eta = \text{softplus}(\tilde{\epsilon}) = \log\{1 + \exp(\tilde{\epsilon})\}, \quad \text{where } \tilde{\epsilon} \sim \mathcal{N}(\mu_0, \sigma_0), \tag{9}$$

where the probability density function of the iSP-Gaussian distribution $\mathcal{N}_{isp}$ (Fig. 1d) is defined as

$$\mathcal{N}_{isp}(x; \mu_0, \sigma_0) = \frac{1}{\sqrt{2\pi}\sigma_0} \frac{\exp(x)}{\exp(x) - 1} \exp\left\{-\frac{1}{2\sigma_0^2}(\log[\exp(x) - 1] - \mu_0)^2\right\}, \tag{10}$$

which is parameterized by $\mu_0$ and $\sigma_0$; the details to derive Eq. 10 are described in Appendix. As shown in Eq. 9, the iSP-Gaussian produces $\eta$ on a positive domain $(0, +\infty)$ as in the half-Gaussian $\mathcal{N}_h$. In the iSP-Gaussian model, the mean and variance are roughly decoupled due to the two parameters of $\mu_0$ and $\sigma_0$; the standard deviation of the iSP-Gaussian is upper-bounded by the parameter $\sigma_0$ even on the larger mean (Fig. 1d) in contrast to the half-Gaussian model.

The *iSP-Gaussian pooling* is thus formulated by applying the iSP-Gaussian distribution in Eq. 9 to the stochastic pooling scheme in Eq. 5 as,

$$Y = \mu_x + \text{softplus}(\mu_0 + \epsilon\sigma_0)\sigma_x, \tag{11}$$

where $\epsilon \sim \mathcal{N}(0, 1)$ and the two variable parameters $\mu_0$ and $\sigma_0$ are estimated by GFGP [1];

$$\mu_0 = \text{f}_\mu(\boldsymbol{X}) = b_\mu + \boldsymbol{v}_\mu^\top \text{ReLU}(\boldsymbol{a} + \boldsymbol{U}^\top \text{GAP}(\boldsymbol{X})), \tag{12}$$

$$\sigma_0 = \text{sigmoid} \circ \text{f}_\sigma(\boldsymbol{X}) = \text{sigmoid}(b_\sigma + \boldsymbol{v}_\sigma^\top \text{ReLU}(\boldsymbol{a} + \boldsymbol{U}^\top \text{GAP}(\boldsymbol{X}))). \tag{13}$$

We employ the same structure of MLP as in Eq. 7 and the first layer ($\boldsymbol{U}$ and $\boldsymbol{a}$) is shared for estimating $\mu_0$ and $\sigma_0$. While the parameter $\mu_0$ can take any value, $\mu_0 \in (-\infty, +\infty)$, the parameter $\sigma_0$ is subject to the non-negativity constraint since those two parameters indicate the mean and standard deviation of the underlying Gaussian distribution in Eq. 9. And, according to the fundamental model in Eq. 2, we further impose the constraint of $\sigma_0 \in (0, 1)$ which also contributes to stable training. It should be noted that even though $\sigma_0$ is so upper-bounded, the variation of the output $Y$ in the stochastic training is proportional to $\sigma_x$ as shown in Eq. 11. Based on these ranges of the parameters, the GFGP model is formulated in Eq. 12 for $\mu_0$ and in Eq. 13 for $\sigma_0$ by applying $\text{sigmoid}(x) = \frac{1}{1+\exp(-x)}$.

The deterministic pooling form at inference is defined by

$$Y = \mu_x + \text{softplus}(\mu_0)\sigma_x, \tag{14}$$

Table 1: Gaussian-based pooling methods. For comparison, the special cases (the deterministic pooling by $\sigma_0 = 0$) of the half-Gaussian and iSP-Gaussian models are shown in the last two rows.

| Pooling form: $Y_{\boldsymbol{q}}^c = \mu_{X,\boldsymbol{q}}^c + \eta_{\boldsymbol{q}}^c \sigma_{X,\boldsymbol{q}}^c$, | | Random number: $\epsilon_{\boldsymbol{q}}^c \sim \mathcal{N}(0,1)$ | |
|---|---|---|---|
| Pooling method | $\eta_{\boldsymbol{q}}^c$ at training | $\eta_{\boldsymbol{q}}^c$ at inference | Parameter |
| Gaussian | $\epsilon_{\boldsymbol{q}}^c$ | $0$ | - |
| Half-Gaussian (fixed) | $\lvert\epsilon_{\boldsymbol{q}}^c\rvert$ | $\frac{\sqrt{2}}{\sqrt{\pi}}$ | - |
| Half-Gaussian | $\lvert\epsilon_{\boldsymbol{q}}^c\rvert\sigma_0^c$ | $\frac{\sqrt{2}}{\sqrt{\pi}}\sigma_0^c$ | $\sigma_0^c = \texttt{softplus} \circ \texttt{f}(\boldsymbol{X})$ |
| iSP-Gaussian | $\texttt{softplus}(\mu_0^c + \epsilon_{\boldsymbol{q}}^c\sigma_0^c)$ | $\texttt{softplus}(\mu_0^c)$ | $\mu_0^c = \texttt{f}_\mu(\boldsymbol{X}),\ \sigma_0^c = \texttt{sigmoid} \circ \texttt{f}_\sigma(\boldsymbol{X})$ |
| Average | | $0$ | - |
| iSP-Gaussian ($\sigma_0 = 0$) | | $\texttt{softplus}(\mu_0^c)$ | $\mu_0^c = \texttt{f}_\mu(\boldsymbol{X})$ |

where $\mu_0 = \texttt{f}_\mu(\boldsymbol{X})$ in Eq. 12 and we approximate the mean of the iSP-Gaussian distribution as

$$\mathrm{E}[\eta] = \int \log[1 + \exp(\tilde{\epsilon})]\mathcal{N}(\tilde{\epsilon}; \mu_0, \sigma_0)d\tilde{\epsilon} \approx \texttt{softplus}(\mu_0) + 0.115\sigma_0^2 \frac{4\exp(0.9\mu_0)}{(1 + \exp(0.9\mu_0))^2} \quad (15)$$

$$\approx \texttt{softplus}(\mu_0). \quad (16)$$

The first approximation in Eq. 15 is given in a heuristic manner[2] for $\sigma_0 \leq 1$ and the second one in Eq. 16 is obtained by ignoring the residual error which is at most 0.115. In the preliminary experiments, we confirmed that the approximation hardly degrades classification performance (at most only 0.01% drop), and it is practically important that the approximation halves the GFGP computation only for $\mu_0 = \texttt{f}_\mu(\boldsymbol{X})$ by omitting $\sigma_0$ in Eq. 16.

## 2.4 Discussion

**Training** The proposed Gaussian-based pooling methods are summarized in Table 1. These methods leverage a random number $\epsilon$ simply drawn from a normal distribution $\mathcal{N}(0,1)$ to the stochastic training which is based on the following derivatives,

$$\frac{\partial Y_{\boldsymbol{q}}^c}{\partial X_{\boldsymbol{p}}^c} = \frac{1}{|\mathcal{R}_{\boldsymbol{q}}|}\left(1 + \eta_{\boldsymbol{q}}^c \frac{X_{\boldsymbol{p}}^c - \mu_{X,\boldsymbol{q}}^c}{\sigma_{X,\boldsymbol{q}}^c}\right), \quad \frac{\partial Y_{\boldsymbol{q}}^c}{\partial \eta_{\boldsymbol{q}}^c} = \sigma_{X,\boldsymbol{q}}^c. \quad (17)$$

While the pooling parameters $\{\mu_0^c, \sigma_0^c\}$ are estimated by GFGP for channels $c \in \{1, \cdots, C\}$, the random number $\epsilon_{\boldsymbol{q}}^c$ is generated at each position $\boldsymbol{q}$ and channel $c$, i.e., for each output $Y_{\boldsymbol{q}}^c$. To reduce the memory consumption in the stochastic training process, it is possible to utilize random numbers $\epsilon^c$ which are generated only along the channel $c$ and shared among spatial positions $\boldsymbol{q}$; this approach is empirically evaluated in Section 3.1.

**iSP-Gaussian model** As an alternative to the iSP-Gaussian, the log-Gaussian model [4] is applicable in Eq. 11 with the analytic form of mean, $\exp(\mu_0 + \frac{\sigma_0^2}{2})$. Nonetheless, the iSP-Gaussian model is preferable for pooling in the following two points. First, the mean of iSP-Gaussian can be approximated by using the single variable $\mu_0$ in Eq. 16 in order to effectively reduce computation cost at inference by omitting the estimation of $\sigma_0$ in the GFGP method. Second, the variance of iSP-Gaussian is upper-bounded by $\sigma_0^2$ for any $\mu_0$, while the log-Gaussian model exponentially enlarges the variance as $\mu_0$ increases, leading to unstable training; in the preliminary experiment, we confirmed that the log-Gaussian model fails to properly reduce the training loss.

**Pooling model** The proposed pooling forms in Table 1 are based on a linear combination of the *average* and *standard deviation* pooling both of which have been practically applied to extract visual characteristics [3, 31]. In the proposed method, those two statistics are fused through the probabilistic model of which parameter(s) is estimated by GFGP [1] from an input feature map. Estimating parameters of a probabilistic model by neural networks is found in the mixture density network (MDN) [2] and partly in variational auto-encoder (VAE) [12]. The proposed method effectively applies the approach to stochastic training of CNN in the framework of stochastic pooling.

**Computation complexity**   In the iSP-Gaussian pooling, the computation overhead is mainly caused by the GFGP module. The GFGP method estimates $2C$ parameters, $\{\mu_0^c, \sigma_0^c\}_{c=1}^C$, by means of two-layered MLP in Eqs. 12 13 equipped with $\frac{3}{2}C^2 + \frac{5}{2}C$ parameters which efficiently performs in $O(C^2)$ due to GAP; the efficiency of GFGP compared to the other methods is shown in [1]. The pooling operation itself in the proposed method is more efficient than [1] since it is composed of two simple statistics, local mean $\mu_X$ and standard deviation $\sigma_X$.

# 3   Experimental Results

We apply the proposed pooling methods (Table 1) to various CNNs on image classification tasks; *local* pooling layers embedded in original CNNs are replaced with our proposed ones. The classification performance is evaluated by error rates (%) on a validation set provided by datasets. The CNNs are implemented by using MatConvNet [26] and trained on NVIDIA Tesla P40 GPU.

## 3.1   Ablation study

To analyze the proposed Gaussian-based pooling methods (Table 1) from various aspects, we embed them in the pool1&2 layers of the 13-layer network (Table 2a) on the Cifar100 dataset [13] which contains 50,000 training images of $32 \times 32$ pixels and 10,000 validation images of 100 object categories; the network is optimized by SGD with a batch size of 100, weight decay of 0.0005, momentum of 0.9 and the learning rate which is initially set to 0.1 and then divided by 10 at the 80th and 120th epochs over 160 training epochs, and all images are pre-processed by standardization (0-mean and 1-std) and for data augmentation, training images are subject to random horizontal flipping and cropping through 4-pixel padding. We repeat the evaluation three times with different initial random seeds in training the CNN to report the averaged error rate with the standard deviation.

**Probabilistic model**   In Section 2, we start with the simple Gaussian model in Eq. 2 and then derive various probabilistic models for pooling, as summarized in Table 1. The performance comparison of those methods are shown in Table 2b where the former four methods are stochastic while the latter two are deterministic. By embedding stochasticity into the local pooling, the performance is improved, and the half-Gaussian model is superior to the simple Gaussian model since it excludes the effect of min-pooling (Fig. 1c) by favorably activating inputs due to non-negative $\eta$ in Eq. 4. Then, the performance is further improved by extending the fixed half-Gaussian model to the more flexible ones through introducing variable pooling parameters to be estimated by GFGP [1]; in this case, the half-Gaussian (Eq. 6) and the iSP-Gaussian (Eq. 11) work comparably. The comparison to the deterministic iSP-Gaussian model ($\sigma_0 = 0$) clarifies that it is quite effective to incorporate stochasticity into GFGP via the prior probabilistic models. The trainable pooling by GFGP could slightly bring an over-fitting issue especially in such a small-scale case, and the proposed stochastic method mitigates the issue to favorably exploit the discriminative power of the GFGP model for improving performance.

**Parametric model**   From the viewpoint of the increased number of parameters, we show the effectiveness of the proposed method in comparison with the other types of modules that adds the same number of parameters; NiN [17] using $1 \times 1$ conv, ResNiN which adds an identity path to the NiN module as in ResNet [7], and squeeze-and-excitation (SE) module [9]. For fair comparison, they are implemented by using the same 2-layer MLP as ours (Eq. 12) of $C^2$ parameters with appropriate activation functions and are embedded before pool1&2 layers in the 13-layer Net (Table 2a) so as to work on the feature map fed into the *max* pooling layer. The performance results are shown in Table 2c, demonstrating that our method most effectively leverages the additional parameters to improve performance.

**Stochastic method**   There are several methods which introduce stochasticity into the convex pooling (Eq. 1); *Stochastic Pooling* [33] constructs a multinomial model on the weights $w_p$ by directly using input activation $X_p$, and *Mixed Pooling* [15] mixes average- and max-pooling in a stochastic manner. Those methods are compared with the proposed methods of the half-Gaussian and iSP-Gaussian models in Table 2d, demonstrating the superiority of the proposed methods to the previous stochastic methods. On the other hand, $S^3$ pooling [34] endows local pooling with stochasticity in a different way from ours and the methods [15, 33]; $S^3$ pooling stochastically selects the receptive field $\mathcal{R}_q$ of the output $Y_q$, and thus can be combined with the above-mentioned methods that consider the stochasticity in producing $Y_q$ based on $\mathcal{R}_q$. As shown in Table 2d, the combination methods with

Table 2: Performance results by 13-layer network (a) on Cifar100 dataset [13].

### (a) 13-layer network

| input | $32 \times 32$ RGB image |
|---|---|
| conv 1a | 96 filters, $3 \times 3$, pad = 1, BatchNorm, ReLU |
| conv 1b | 96 filters, $3 \times 3$, pad = 1, BatchNorm, ReLU |
| conv 1c | 96 filters, $3 \times 3$, pad = 1, BatchNorm, ReLU |
| pool1 | Pooling, $2 \times 2$, pad = 0 |
| conv 2a | 192 filters, $3 \times 3$, pad = 1, BatchNorm, ReLU |
| conv 2b | 192 filters, $3 \times 3$, pad = 1, BatchNorm, ReLU |
| conv 2c | 192 filters, $3 \times 3$, pad = 1, BatchNorm, ReLU |
| pool2 | Pooling, $2 \times 2$, pad = 0 |
| conv 3a | 192 filters, $3 \times 3$, pad = 1, BatchNorm, ReLU |
| conv 3b | 192 filters, $3 \times 3$, pad = 1, BatchNorm, ReLU |
| conv 3c | 192 filters, $3 \times 3$, pad = 1, BatchNorm, ReLU |
| GAP | Global average-pooling (GAP), $8 \times 8 \to 1 \times 1$ |
| dense | Fully connected, $192 \to 100$ |
| output | Softmax |

### (d) Stochastic method

| Method | Error (%) |
|---|---|
| Stochastic [33] | $24.52 \pm 0.18$ |
| Mixed [15] | $24.33 \pm 0.23$ |
| Half-Gauss | $23.48 \pm 0.22$ |
| iSP-Gauss | $23.52 \pm 0.37$ |
| $S^3$ [34] + Stochastic [33] | $24.01 \pm 0.20$ |
| $S^3$ [34] + Mixed [15] | $23.31 \pm 0.12$ |
| $S^3$ [34] + Half-Gauss | $23.12 \pm 0.17$ |
| $S^3$ [34] + iSP-Gauss | $22.98 \pm 0.02$ |

### (b) Probabilistic model

| Method | Error (%) |
|---|---|
| Gaussian | $24.51 \pm 0.36$ |
| Half-Gauss (fixed) | $24.25 \pm 0.25$ |
| Half-Gauss | $23.48 \pm 0.22$ |
| iSP-Gauss | $23.52 \pm 0.37$ |
| Average | $24.78 \pm 0.18$ |
| iSP-Gauss ($\sigma_0 = 0$) | $24.12 \pm 0.17$ |

### (c) Parametric model

| Method | Error (%) |
|---|---|
| NiN [17] | $24.49 \pm 0.13$ |
| ResNiN [7, 17] | $24.33 \pm 0.16$ |
| SE [9] | $23.99 \pm 0.07$ |
| iSP-Gauss | $23.52 \pm 0.37$ |

### (e) Global pooling

| Method | Error (%) |
|---|---|
| GAP | $24.78 \pm 0.18$ |
| GAP + DropOut [16] | $24.58 \pm 0.27$ |
| Half-Gauss | $24.54 \pm 0.14$ |
| iSP-Gauss | $23.83 \pm 0.18$ |

### (f) Stochasticity

| Method | Full ($\epsilon_q^c$) | Partial ($\epsilon^c$) |
|---|---|---|
| Half-Gauss | $23.48 \pm 0.22$ | $23.60 \pm 0.07$ |
| iSP-Gauss | $23.52 \pm 0.37$ | $23.68 \pm 0.06$ |

the $S^3$ pooling [34] favorably improve performance. The half-Gaussian model, however, enjoys the smaller amount of improvement, compared to the iSP-Gaussian model. The half-Gaussian model provides higher stochasticity by nature due to the large variance (Fig. 1c), which might make the additional stochasticity by $S^3$ less effective.

**Global pooling**  While in this paper we focus on the operation of *local* pooling in CNNs, it is possible to apply the proposed method to globally aggregate features after the last convolution layer as the global average pooling (GAP) does. To evaluate the feasibility to global pooling, we replace the GAP with the proposed pooling methods in the 13-layer network (Table 2a) which is equipped with local average pooling. For comparison in terms of stochasticity, we also apply DropOut [24] to GAP; as suggested in [16], the DropOut layer with the dropping ratio 0.2 is embedded just after the GAP so as to achieve performance improvement for the *batch-normalized* CNNs. The performance comparison is shown in Table 2e, and we can see that the iSP-Gaussian pooling effectively works in the global pooling. On the other hand, the half-Gaussian model is less effective, maybe due to its higher stochasticity as pointed out above; the global pooling would require small amount of stochasticity as implied by the result that the DropOut with the ratio 0.2 works [16]. And, we can note that the DropOut operating on the last layer [16] is compatible with the *local* pooling methods.

**Stochasticity**  Full stochastic training is realized by performing stochastic sampling at each output neuron $Y_q^c$ individually, i.e., by drawing the random number $\epsilon_q^c$ for each $\{q, c\}$ (Table 1). Such a full stochastic approach, however, requires considerable amount of memory and computation cost for $\epsilon_q^c$ especially on the larger-sized input images, as mentioned in Section 2.4. To increase computation efficiency in training, we can apply partially stochastic training only along the channels $c$; that is, all the neurons $\{Y_q^c\}_q$ on the $c$-th channel map share the identical $\epsilon^c$ which is sampled from a normal distribution in a channel-wise manner. It is noteworthy that even in this partially stochastic scheme the output $Y_q^c$ is differently distributed based on $\mu_{X,q}^c$ and $\sigma_{X,q}^c$ computed at each $q$. These two types of stochastic schemes are compared in Table 2f. The partially stochastic approach produces favorable performance, though slightly degrading performance. Thus, we apply this computationally efficient stochastic approach to the larger CNN models on ImageNet dataset in Section 3.2.

## 3.2  Comparison to the other pooling methods

Next, the proposed pooling methods of the half-Gaussian and iSP-Gaussian models are compared to the other local pooling methods on various CNNs. For comparison, in addition to the stochastic

Table 3: Performance comparison on various CNNs.

| Cifar100 [13] (a) 13-layer Net (Table 2a) | | (b) MobileNet [8] | | ImageNet [5] (c) ResNet-50 [7] | | | (d) ResNeXt-50 [30] | | | |
|---|---|---|---|---|---|---|---|---|---|---|
| Method | Error (%) | Method | Top-1 | Top-5 | Method | Top-1 | Top-5 | Method | Top-1 | Top-5 |
| skip | $24.83 \pm 0.15$ | skip | 29.84 | 10.35 | skip | 23.53 | 7.00 | skip | 22.69 | 6.65 |
| avg | $24.78 \pm 0.18$ | avg | 28.94 | 10.00 | avg | 22.61 | 6.52 | avg | 22.14 | 6.35 |
| max | $24.74 \pm 0.08$ | max | 29.23 | 10.02 | max | 22.99 | 6.71 | max | 22.20 | 6.24 |
| Stochastic [33] | $24.52 \pm 0.18$ | Stochastic | 30.26 | 10.64 | stochastic | 25.47 | 7.87 | stochastic | 25.02 | 7.73 |
| Mixed [15] | $24.33 \pm 0.23$ | Mixed | 29.49 | 10.14 | Mixed | 22.81 | 6.53 | Mixed | 21.83 | 6.09 |
| DPP [22] | $24.59 \pm 0.15$ | DPP | 28.92 | 9.92 | DPP | 22.52 | 6.63 | DPP | 21.84 | 5.98 |
| Gated [15] | $24.42 \pm 0.45$ | Gated | 28.62 | 9.86 | Gated | 22.27 | 6.33 | Gated | 21.63 | 5.99 |
| GFGP [1] | $24.41 \pm 0.22$ | GFGP | 27.68 | 9.27 | GFGP | 21.79 | 5.95 | GFGP | 21.35 | 5.74 |
| Half-Gauss | $23.48 \pm 0.22$ | Half-Gauss | 27.96 | 9.38 | Half-Gauss | 21.66 | 5.88 | Half-Gauss | 20.89 | 5.72 |
| iSP-Gauss | $23.52 \pm 0.37$ | iSP-Gauss | 27.33 | 9.00 | iSP-Gauss | 21.37 | 5.68 | iSP-Gauss | 20.66 | 5.60 |

pooling methods [33, 15], we apply the deterministic pooling methods including the simple average- and max-pooling as well as the sophisticated ones [1, 15, 22] which are trainable in the end-to-end learning. As to CNNs, besides the simple 13-layer network (Table 2a) on the Cifar100 dataset, we train the deeper CNNs of MobileNet [8], ResNet-50 [7] and ResNeXt-50 [30] on the ImageNet dataset [5]; for ResNet-based models, we apply the batch size of 256 to SGD with momentum of 0.9, weight decay of 0.0001 and the learning rate which starts from 0.1 and is divided by 10 every 30 epochs throughout 100 training epochs, while we apply the similar procedure to train the MobileNet over 120 training epochs with the data augmentation of slightly less variation as suggested in [8]. Those deep CNNs contain five local pooling layers in total, including *skip* one implemented by strided convolution, and they are replaced by the other local pooling methods as in [1]. The performance is measured by top-1 and top-5 error rates via single crop testing [14] on the validation set.

The performance comparison in Table 3 shows that the proposed methods favorably improve performance, being superior both to the stochastic pooling methods and to the sophisticated deterministic methods. Thus, we can say that it is effective to fuse the effective deterministic approach via GFGP [1] and the stochastic scheme through the probabilistic model on the local pooling. While the half-Gaussian and iSP-Gaussian models are comparable in the smaller-scale case (Table 3a), the iSP-Gaussian pooling produces superior performance on the larger-scale cases (Table 3b-d). The iSP-Gaussian model that renders appropriate stochasticity through flexibly controlling $\sigma_0$ in Eq. 13 contributes effectively to improving performance of various CNNs.

### 3.3 Qualitative analysis

Finally, we show how the pooling parameters of the iSP-Gaussian model are estimated by GFGP. The model contains two parameters of $\mu_0^c$ and $\sigma_0^c$ at each channel $c$ which are estimated for each input image sample. Fig. 2 visualizes as 2-D histograms the distributions of the parameter pairs $\{\mu_0, \sigma_0\}$ estimated on training samples. At the beginning of the training, the parameters are estimated less informatively, being distributed broadly especially in $\sigma_0$. As the training proceeds, the probabilistic model in the pooling is optimized, and the parameter $\sigma_0$ that controls the stochasticity in training is adaptively tuned at respective layers; we can find some modes in the first two layers of ResNet-50 while in the third and fourth layers $\sigma_0$ slightly exhibits negative correlation with $\mu_0$, suppressing stochasticity on the significant output of high $\mu_0$. By flexibly tuning the model parameters throughout the training, the proposed iSP-Gaussian pooling effectively contributes to improving performance on various CNNs.

## 4   Conclusion

In this paper, we have proposed a novel pooling method based on the Gaussian-based probabilistic model over the local neuron activations. In contrast to the previous pooling model based on the convex hull of local samples (activations), the proposed method is formulated by means of the probabilistic model suitable for pooling functionality in CNNs; we propose the inverse softplus-Gaussian model for

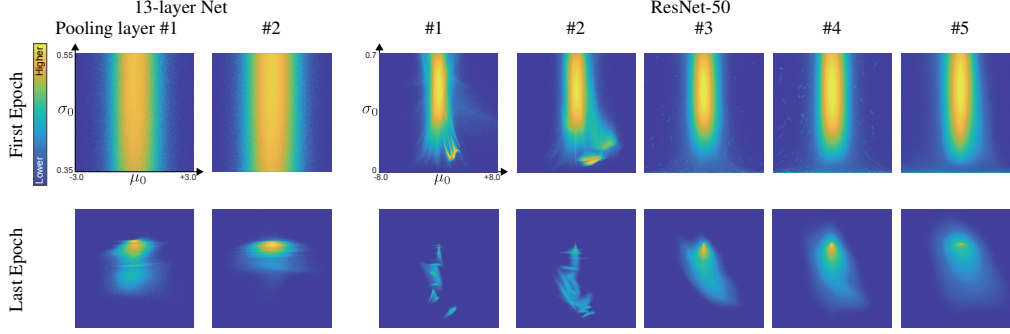

Figure 2: Distribution of the estimated parameters $\mu_0$ and $\sigma_0$ in the iSP-Gaussian model. To construct the 2-D histograms of which frequencies are depicted by pseudo colors, all the training samples of Cifar100 dataset are fed into the 13-layer Net, while in the ResNet-50 we randomly draw 200,000 training samples from ImageNet. This figure is best viewed in color.

that purpose. The local neuron activations are aggregated into the local statistics of mean and standard deviation of the Gaussian model which are then fed into the probabilistic model for performing local pooling stochastically. For controlling the pooling form as well as the stochastic training, the model contains variable parameters to be adaptively estimated by the GFGP method [1]. Thus the proposed method naturally unifies the two schemes of stochastic pooling and trainable pooling. In the experiments on image classification, the proposed method is applied to various CNNs, producing favorable performance in comparison with the other pooling methods.

## Appendix: Derivation of Inverse softplus-Gaussian Distribution $\mathcal{N}_{isp}$

The probability distribution $\mathcal{N}_{isp}(x; \mu_0, \sigma_0)$ in Eq. 10 is derived through the following variable transformation. Suppose $y$ is a random variable whose probability density function is Gaussian,

$$\mathsf{q}(y) = \frac{1}{\sqrt{2\pi}\sigma_0} \exp \left\{ -\frac{1}{2\sigma_0^2} (y - \mu_0)^2 \right\}. \tag{18}$$

The target random variable $x$ is obtained via softplus transformation by

$$x = \texttt{softplus}(y) \Leftrightarrow y = \texttt{softplus}^{-1}(x) = \log[\exp(x) - 1]. \tag{19}$$

Then, we apply the relationship of

$$\mathsf{q}(y)dy = \mathsf{p}(x)dx, \quad \frac{dy}{dx} = \frac{\exp(x)}{\exp(x) - 1} \tag{20}$$

to provide $\mathsf{p}(x) = \mathcal{N}_{isp}(x; \mu_0, \sigma_0)$ in Eq. 10.

## Footnotes

[1]As in *log*-Gaussian distribution [4], *inverse softplus*-Gaussian is a distribution of random variable which is transformed via an inverse softplus function into the variable that obeys a Gaussian distribution.

[2]We *manually* tune the parametric form in Eq. 15 toward minimizing the residual error between $\texttt{softplus}(\mu_0)$ and $\int \log[1 + \exp(\tilde{\epsilon})]\mathcal{N}(\tilde{\epsilon}; \mu_0, \sigma_0)d\tilde{\epsilon}$ which is empirically computed by means of sampling.

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
