[Supplementary Material 1]

# Global Feature Guided Local Pooling

## Abstract

*In deep convolutional neural networks (CNNs), local pooling operation is a key building block to effectively downsize feature maps for reducing computation cost as well as increasing robustness against input variation. There are several types of pooling operation, such as average/max-pooling, from which one has to be manually selected for building CNNs. The optimal pooling type would be dependent on characteristics of features in CNNs and classification tasks, making it hard to find out the proper pooling module in advance. In this paper, we propose a flexible pooling method which adaptively tunes the pooling functionality based on input features without manually fixing it beforehand. In the proposed method, the parameterized pooling form is derived from a probabilistic perspective to flexibly represent various types of pooling and then the parameters are estimated by means of global statistics in the input feature map. Thus, the proposed local pooling guided by global features effectively works in the CNNs trained in an end-to-end manner. The experimental results on large-scale image classification tasks demonstrate that the proposed pooling method produces favorable performance in various deep CNNs.*

## 1. Introduction

Deep convolutional neural networks (CNNs) are successful models for high-performance image recognition [18, 32, 12]. While some techniques such as Batch-Normalization [16] and DropOut [33] are fundamental to properly train the deep models, from the architectural viewpoint, the CNNs mainly comprise three basic operations of convolution, activation and spatial pooling. The activation functions, especially rectified-linear unit (ReLU) [26] and its variants [24, 11], are applied to non-linearly transform the neuron responses. The convolution is a crucial operation to extract effective features from an input image by the learned filters [43, 4, 40]. The *local* convolution is expanded to fully-connected (FC) one [18] which performs *globally*.

In a similar way to the convolution, there are two types of spatial pooling in terms of receptive field, *local* or *global* ones. The *global* pooling effectively substitutes for the FC in some CNNs [22, 12, 34] through spatially compressing the (last) feature map of space-channel tensor into the feature vector of channel dimensionality which is finally fed into classification layers. Practically speaking, the average-pooling is mainly applied to globally aggregate features, though some pooling forms are also investigated in [17].

In contrast to the global pooling, *local* pooling operation is a key building block commonly employed in most CNNs to efficiently reduce spatial resolution with increasing robustness against variations in input images, such as translation. The local pooling also stems from the biological insight [15]. According to the biologically-inspired model [31], various types of deep CNNs [18, 32, 34] employ local max-pooling for downsizing the feature maps, while average-pooling is also applied to CNNs [20]. On the other hand, some models [12, 40] achieve the same effect of downsizing by means of *strided* convolution which is also regarded as a pooling following the convolution of 1-striding [45]. Therefore, in contrast to convolution operation, there are several ways, such as avg/max, to implement the pooling operation and it is hard to manually choose the optimal pooling type; it is determined based on the empirical performance, requiring huge amounts of effort in a trial and error approach. Thus, it motivates us to optimize the type of pooling function in an end-to-end training, as is done for convolution filters or leaky-ReLU parameters [11].

Toward trainable pooling operation, it is necessary to address two issues regarding (1) how to formulate various types of local pooling function and (2) what kind of data to use for determining the pooling type. Diverse pooling functionality has to be represented by a simple and unified form, and the pooling operation in the form should be adaptively tuned based on the input features even on a test phase, since the optimal pooling type is related to the characteristics of input features [1]. Thus, the trainable pooling demands such a dynamic and flexible formulation. In addition, the optimal pooling functionality would be determined based on the *global* characteristics of input features beyond *local* receptive field. The local and global pooling have been discussed separately as above, and there is no fusion between them; the local pooling function so far deals with only local features in the receptive field [30, 21].

In this work, we propose a novel trainable *local* pooling function guided by the *global* features beyond the local

ones. We first formulate the pooling function based on the maximum entropy principle [25] to flexibly represent various types of pooling functions with trainable parameters. The type of pooling functionality is effectively controlled by the parameters. Then, we leverage global feature statistics to estimate the parameters of the flexible pooling function. In the proposed pooling method, the parameters are not directly trained in a static form but dynamically determined by means of the global features, in contrast to the other parametric (trainable) pooling methods [30, 21]. Our main contributions are three folds: (1) we theoretically derive the parameterized pooling function which flexibly describes various types of pooling, (2) incorporate the global features into the pooling operation through adaptively estimating the pooling parameters, and (3) perform thorough experiments on large-scale datasets to present the effective pooling form by analyzing the method from various aspects, while showing the favorable performance in comparison with the other methods.

## 2. Related Works

The spatial pooling originates from the biological work about complex cells in the mammalian visual cortex [15]. Then, the importance of max-pooling has been discussed in some neuroscientific works through analyzing/mimicking the primary visual cortex area V1 [28, 29, 31].

The pooling is also applied for rather practical purpose to aggregate features locally to build the local image descriptors, such as SIFT [23] and HOG [5], via average-pooling. In the framework of bag-of-words [3], the spatial pooling is applied globally to aggregate word codes assigned to the local descriptors toward effective image representation. While average-pooling is widely applied to count words, max-pooling also works well in conjunction with sparse coding techniques [1, 41].

In the literature of neural networks, a pooling technique is applied either to summarize neuron activations across *channel* in a multi-layered perceptron (MLP) or to downsize *spatial* resolution in a convolutional neural network (CNN). In the *channel* pooling, $L_p$-norm is discussed from the viewpoint of signal recovery [2] and is extended to the trainable one through learning the parameter $p$ [9] to smoothly transit from average ($p = 1$) to max ($p = \infty$) operation. The trainable $L_p$ pooling is also related to MAX-OUT [7] which performs inter-channel max-pooling.

The local *spatial* pooling is widely applied to deep CNNs for gradually downsizing the feature maps with increasing computation efficiency and robustness. While the average pooling is employed in some CNNs [20], the max pooling becomes a popular choice for deep CNNs [18, 32, 34] according to the biologically-inspired model of HMAX [31]. Nonetheless, in building CNNs, it is necessary to determine the type of pooling, average or max, and the opti-

mal pooling type would be dependent on the recognition tasks and/or characteristics of the input features [1]. There are works [42, 21, 30] to address the problem by filling the gap between these two pooling operations, with similar motivation to ours. The average and max pooling functions can be simply integrated in a convex combination with one weighting parameter which is randomly selected in [42] or trained based on the local features by [21]. To mitigate the artifacts caused by downsizing, the image processing techniques are also applied to the pooling [39, 30]. Wavelet pooling [39] employs the wavelet to compress a feature map accurately with less artifacts. The detailed preserving pooling (DPP) [30] is also proposed based on the image downscaling technique [38] to formulate a parametric pooling function as an intermediate operation between average and max. The proposed method is close to those methods [21, 30] but is clearly different in the following two points; the parametric pooling function is derived in a theoretical way based on maximum entropy principle [25] to flexibly represent various types of pooling including average/max-pooling (Sec. 3), and the parameters are dynamically estimated from the global features beyond the local receptive field of the pooling (Sec. 4).

From the probabilistic viewpoint, stochastic approaches are applied to the local spatial pooling function [44, 45, 42] for introducing randomness into the CNNs as in DropOut [33]. In contrast to the deterministic pooling mentioned above, the stochastic methods randomly pick up a neuron activation in the receptive field throughout an end-to-end training. We also discuss the connection between the proposed method and the stochastic pooling in Sec. 3.3.

## 3. Parameterized Pooling Function

The local spatial pooling is generally formulated as follows. Given an input feature map $\boldsymbol{X} \in \mathbb{R}^{H \times W \times C}$, the $c$-th channel output $Y_p^c$ at the pixel $p$ is computed by applying the spatial pooling to the $c$-th channel feature map as

$$Y_p^c = \sum_{q \in \mathcal{R}_p} \mathbb{W}_p^c(q, \boldsymbol{X}) X_q^c, \qquad (1)$$

$$s.t. \ \mathbb{W}_p^c(q, \boldsymbol{X}) \geq 0, \ \forall q \in \mathcal{R}_p, \ \sum_{q \in \mathcal{R}_p} \mathbb{W}_p^c(q, \boldsymbol{X}) = 1, \ \forall p, c,$$

where $\mathcal{R}_p$ is a set of pixel positions in the receptive field (neighborhood) of the pixel $p$, and $\mathbb{W}_p^c$ is a weighting function to represent the type of pooling. For example, *average-*pooling [20] is realized by $\mathbb{W}_p^c(q, \boldsymbol{X}) = \frac{1}{|\mathcal{R}_p|}$ in disregard of both the position $q$ and the features $\boldsymbol{X}$, and *max-*pooling [32] is represented by $\mathbb{W}_p^c(q^*, \boldsymbol{X}) = 1, \mathbb{W}_p^c(q \neq q^*, \boldsymbol{X}) = 0$ where $q^* = \arg\max_{q \in \mathcal{R}_p} X_q^c$, while *skip-ping* [12] is simply given by $\mathbb{W}_p^c(q = p, \boldsymbol{X}) = 1, \mathbb{W}_p^c(q \neq p, \boldsymbol{X}) = 0$. Thus, designing pooling operation results in how to formulate the weighting function $\mathbb{W}_p^c$.

## 3.1. Maximum entropy principle

In this work, based on the constraints in (1), we regard the weighing function $\mathtt{W}_p^c$ as a probability density function, and from the probabilistic perspective, formulate the following optimization problem for $\mathtt{W}_p^c$:

$$\max_{\mathtt{W}} \sum_{q \in \mathcal{R}_p} -\mathtt{W}(q) \log[\mathtt{W}(q)] + \tilde{\lambda}\mathtt{W}(q)X_q - \eta\mathtt{W}(q) \log\left[\frac{\mathtt{W}(q)}{\tilde{\rho}_q}\right],$$

$$(2)$$

$$s.t. \ \mathtt{W}(q) \geq 0, \ \sum_{q \in \mathcal{R}_p} \mathtt{W}(q) = 1, \tag{3}$$

where we omit in $\mathtt{W}$ the notations of $p$, $c$ and $\boldsymbol{X}$ for simplicity and introduce regularization parameters $\tilde{\lambda} \in \mathbb{R}$ and $\eta > 0$ as well as position prior distribution $\{\tilde{\rho}_q\}_{q \in \mathcal{R}_p}$. The first term of (2) is derived from the maximum entropy principle [25] which is a natural assumption to determine the probability densities. In addition, two regularizations regarding the output $Y_p^c$ (1) and the position are introduced in the second and the third terms, respectively. The second term intends to make the output $Y = \mathtt{W}(q)X_q$ more distinctive via maximization ($\tilde{\lambda} > 0$) or minimization ($\tilde{\lambda} < 0$). And, the weighing $\mathtt{W}$ should be close to the predefined position priors $\{\tilde{\rho}_q\}$ via minimizing Kullback–Leibler divergence in the third term.

(2) can be solved by introducing the Lagrange multipliers $\alpha_q \geq 0$ and $\beta \in \mathbb{R}$ for the non-netativity and unit-sum constraints in (3) to provide the derivative w.r.t. $\mathtt{W}(q)$ as

$$-(1+\eta)(1+\log[\mathtt{W}(q)]) + \tilde{\lambda}X_q + \eta \log[\tilde{\rho}_q] + \alpha_q + \beta = 0, \quad (4)$$

which leads to the following form of $\mathtt{W}$,

$$\mathtt{W}(q) = \exp\left\{\frac{1}{1+\eta}(\tilde{\lambda}X_q + \eta \log[\tilde{\rho}_q] + \alpha_q + \beta) - 1\right\} \quad (5)$$

$$= \frac{\exp(\lambda X_q + \rho_q)}{\sum_{q' \in \mathcal{R}_p} \exp(\lambda X_{q'} + \rho_{q'})}, \tag{6}$$

where $\alpha_q = 0$ due to the positivity of (5) and the KKT condition, and $\beta$ is defined so as to satisfy the unit-sum constraint. We also reparameterize $\lambda = \frac{\tilde{\lambda}}{1+\eta} \in \mathbb{R}$ and $\rho_q = \frac{\eta}{1+\eta} \log[\tilde{\rho}_q] + \epsilon$ with some constant $\epsilon$ to let $\rho_q \in \mathbb{R}$ without loss of generality. The weighting function $\mathtt{W}$ is decomposed into $\mathtt{W}(q) \propto \exp(\lambda X_q) \exp(\rho_q)$ which comprises two kinds of weights regarding feature $X$ and spatial position $q$, as in the bilateral filter [35]. Finally, we obtain the parameterized pooling function by

$$Y_p^c = \frac{\sum_{q \in \mathcal{R}_p} X_q^c \exp(\lambda^c X_q^c + \rho_{q-p}^c)}{\sum_{q \in \mathcal{R}_p} \exp(\lambda^c X_q^c + \rho_{q-p}^c)}, \tag{7}$$

where we make the parameters independent of the position $p$ and dependent only on the channel $c$. Namely, the pooling function contains the parameters of $\boldsymbol{\lambda} = \{\lambda^c\}_{c=1}^C \in \mathbb{R}^C$ and $\boldsymbol{\rho} = \{\rho_{q-p}^c\}_{c=1, q \in \mathcal{R}_p}^C \in \mathbb{R}^{C|\mathcal{R}_p|}$ where '$q$–$p$' means the relative position from $p$ in the receptive field $\mathcal{R}_p$. Thereby, the parameters are shared across any position $p$ at that layer.

## 3.2. Derivative

The derivatives of the pooling function (7) is given by

$$\frac{\partial Y_p^c}{\partial X_q^c} = \frac{\exp(\lambda^c X_q^c + \rho_{q-p}^c)}{\sum_{q' \in \mathcal{R}_p} \exp(\lambda^c X_{q'}^c + \rho_{q'-p}^c)}\left\{1 + \lambda(X_q^c - Y_p^c)\right\},$$

$$(8)$$

$$\frac{\partial Y_p^c}{\partial \lambda} = \frac{\sum_{q \in \mathcal{R}_p} (X_q^c - Y_p^c)^2 \exp(\lambda X_q^c + \rho_{q-p}^c)}{\sum_{q' \in \mathcal{R}_p} \exp(\lambda^c X_{q'}^c + \rho_{q'-p}^c)}, \tag{9}$$

$$\frac{\partial Y_p^c}{\partial \rho_q} = \frac{(X_q^c - Y_p^c) \exp(\lambda X_q^c + \rho_{q-p}^c)}{\sum_{q' \in \mathcal{R}_p} \exp(\lambda^c X_{q'}^c + \rho_{q'-p}^c)}, \tag{10}$$

where we consider the derivatives w.r.t. the input feature $X_q^c$ as well as the two parameters $\lambda^c$ and $\rho_{q-p}^c$, all of which can be trained through back-propagation in an end-to-end manner. The parameter $\lambda^c$ is updated by (9) based on the variance of the features, adapting to the scale of the features. On the other hand, the derivative (8) is summed up to 1 on the receptive field $q \in \mathcal{R}_p$, as is the case with the standard pooling such as avg/max-pooling. The update is distributed over the receptive field according to the contribution measured by $\frac{\exp(\lambda^c X_q^c + \rho_{q-p}^c)}{\sum_{q' \in \mathcal{R}_p} \exp(\lambda^c X_{q'}^c + \rho_{q'-p}^c)}$, suppressing the features that are significantly smaller than the average $Y_p^c$, as in *Lateral inhibition* [8], to let the network extract diverse features.

## 3.3. Discussion

**Flexibility.** The pooling function (7) parameterized by $\boldsymbol{\lambda}$ and $\boldsymbol{\rho}$ flexibly describes various types of pooling functions; *average* and *max* pooling are produced by $\{\lambda = 0, \rho_{q-p} = 0\}$ and $\{\lambda \to \infty, \rho_{q-p} = 0\}$, respectively, while $\{\lambda = 0, \rho_{q-p} = \delta_{q-p}\}$ leads to *skipping*. In addition, we can also realize *min*-pooling by $\{\lambda \to -\infty, \rho_{q-p} = 0\}$. The position prior, which endows the pooling with local position sensitivity, can also be related to a wavelet filter in the Wavelet pooling [39]. Such a flexibility of the parametric pooling (7) is a key property for adaptively controlling the pooling type via the global features (Sec. 4).

**Softmax.** Without position priors, *i.e.*, $\boldsymbol{\rho} = \mathbf{0}$, the pooling (7) corresponds to the $\alpha$-softmax [19], which is mentioned in the literature of neuroscience [29, 27] and in the framework of bag-of-words [1]. And, the parameter $\lambda$ is connected to the scaling factor in $L_2$-normalized softmax [37] and to the temperature of softmax in the other literatures [13, 10]; $\lambda$ is a reciprocal of the temperature, $\lambda = \frac{1}{T}$. In those methods, the temperature is tuned by hand to properly transfer the network structure in [13] and

is discriminatively learned based on labeled data for fine-tuning the confidence of the classifier in [10] or for improving the classification performance in the $l_2$-normalized softmax [37]. In contrast, we naturally derive the parameter $\lambda$ from the optimization problem (2) based on the maximum entropy principle. Through controlling the two terms of the entropy and the significance of the output in (2), the parameter $\lambda$ plays a role in smoothly switching the pooling functionality from average to extreme ones (min/max). In addition, the proposed method is distinctive in that the pooling parameters $\boldsymbol{\lambda}$ and $\boldsymbol{\rho}$ in (7) are adaptively determined by means of global features as described in Sec. 4.

**Probabilistic viewpoint.** As described in Sec. 3.1, the pooling (7) outputs the probabilistic mean of the local features according to the probability density function W in (6). On the other hand, the stochastic pooling [44] picks up the feature from the receptive field $\{X_q^c\}_{q \in \mathcal{R}_p}$ according to the probability proportional to the non-negative feature values, $p_q \propto X_q^c$, which is related to (6). For small $\lambda^c$ and $\boldsymbol{\rho} = \mathbf{0}$, we can approximate $\exp(\lambda^c X_q^c) \approx 1 + \lambda^c X_q^c$, and thereby the pooling weights (6) result in the form of $\mathtt{W}(q) \approx \frac{1 + \lambda^c X_q^c}{|\mathcal{R}_p| + \lambda^c \sum_{q' \in \mathcal{R}_p} X_{q'}^c}$ corresponding to the biased probability of [44]. And, we can say that the form (6) is more flexible since it is applicable to any features while the stochastic pooling [44] accepts only non-negative features produced such as by ReLU.

## 4. Global Feature Guidance

We leverage the *global* features to estimate the *parameters* of $\boldsymbol{\lambda}$ and $\boldsymbol{\rho}$ which control the type of pooling function in (7) operating *locally* on the receptive field $\{X_q^c\}_{q \in \mathcal{R}_p}$. For that purpose, the pooling parameters $\boldsymbol{\lambda}$ and $\boldsymbol{\rho}$ are regarded as *variables* rather than static *parameters* to be optimized in the training. We let the variables be dependent on the input features $\boldsymbol{X} \in \mathbb{R}^{H \times W \times C}$ and thus described by the mapping $\boldsymbol{\lambda} = \mathtt{f}(\boldsymbol{X})$ and $\boldsymbol{\rho} = \mathtt{g}(\boldsymbol{X})$. Following the methodology in squeeze-and-excitation (SE) [14], the mapping functions $\mathtt{f}$ and $\mathtt{g}$ are designed by means of the multilayer perceptron (MLP) applied to the global feature statistics (Fig. 1);

$$\boldsymbol{\lambda} = \mathtt{f}(\boldsymbol{X}; \boldsymbol{U}, \boldsymbol{V}_\lambda) \tag{11}$$
$$= \mathtt{s}\left(\boldsymbol{V}_\lambda \, \mathtt{ReLU}(\boldsymbol{U} \, [\mathtt{t}(\{X_p^1\}_{\forall p}), \cdots, \mathtt{t}(\{X_p^C\}_{\forall p})]^\top)\right), \tag{12}$$

where we consider one hidden layer of $D$ neurons with ReLU and the function $\mathtt{t}$ computes $k$ types of statistics per channel over the inputs $\{X_p^c\}_{p=(1,1)}^{(H,W)}$ and $\mathtt{s}$ is an element-wise non-linear activation function; for example, we can set $\mathtt{t}$ to a global averaging function ($k = 1$) and $\mathtt{s}$ to a sigmoid function as in [14]. The similar MLP is applied for $\boldsymbol{\rho}$ by $\mathtt{g}(\boldsymbol{X}; \boldsymbol{U}, \boldsymbol{V}_\rho)$ which shares $\boldsymbol{U}$ with $\mathtt{f}$, though the activation $\mathtt{s}$ may be different. Thus, in the proposed pooling, called *global feature guided pooling* (**GFGP**), the MLP weights

Figure 1. Global Feature Guided Pooling (GFGP).

$\boldsymbol{U} \in \mathbb{R}^{D \times kC}, \boldsymbol{V}_\lambda \in \mathbb{R}^{C \times D}$ and $\boldsymbol{V}_\rho \in \mathbb{R}^{|\mathcal{R}_p|C \times D}$ are the targets to be optimized in an end-to-end training.

In the proposed method, the key point is to deal with the pooling parameters as **variables** to be mapped from the global features by (11). Thus, from that viewpoint, it contrasts with the other methods, as follows.

**Constant.** The pooling with pre-fixed constant $\lambda$ produces such as avg-pooling ($\lambda = 0$), max-pooling ($\lambda = \infty$) and the intermediate one between them [1] ($0 < \lambda < \infty$). In contrast to those pre-fixed pooling, the proposed method determines the type of pooling adaptively based on data without manually tuning it, as in the parameterized pooling [30, 21].

**Parameter.** In [30], the pooling parameters are directly trained in an end-to-end manner. Thus, the trained pooling types could be varied across both channels and layers adaptively unlike the above-mentioned constant pooling. However, the pooling types, *i.e.*, pooling parameters, are fixed once trained, for a test phase. On the other hand, the proposed pooling is dynamically dependent on the input features $\boldsymbol{X}$ via the mapping (11); that is, the same pooling layer works differently according to the input features, which exhibits distinctiveness compared to the previous parameterized pooling [30].

**Locality.** While the gated pooling [21] also estimates the gating parameter via the features yet *locally*, the functionality of the proposed pooling is determined based on the *global* characteristics of input features and works in the *local* receptive field. Thus, we can say that the proposed method incorporates both *local* and *global* information into the pooling. The pooling is generally applied to downsize the spatial resolution of an input feature map, inevitably loosing (spatial) information, and the global information in the proposed pooling would compensate it for improving performance.

## 5. Experimental Results

We evaluate the performance of the proposed method by embedding it into the deep CNNs trained on large-scale datasets for image classification [6, 47]. The classification

performance is measured by single-crop top-1 and top-5 error rates (%) on the validation set of dataset.

## 5.1. Ablation study

We analyze the proposed pooling method (7, 11) on ImageNet classification [6]. It is applied to the deep CNN of VGG-13 [32] which is slightly modified from the original model [32] by introducing Batch-Normalization [16] and reducing the number of channels in the FC layers from 4096 to 2048. The CNN contains *five* local max-pooling layers with the pool size of $2\times2$ and $(2,2)$-striding to downsize the feature map resolution by a factor of 2. We replace all the local pooling layers by the proposed method and train the CNN by following the learning protocol provided in MatConvNet [36]. Unless otherwise noted, the proposed pooling method is implemented by the default setting (Fig. 1) which applies the sigmoid s and the global average pooling $\mathsf{t}(\{X_p^c\}) = \frac{1}{HW}\sum_{p=(1,1)}^{(H,W)} X_p^c$ to the MLP of $D = \frac{C}{2}$ without position priors $\boldsymbol{\rho} = \mathbf{0}$; these are discussed in Sec. 5.1.2.

### 5.1.1  Global feature guiding model

We clarify how the proposed scheme of *global feature guidance* (Sec. 4) contributes to improving the pooling. As mentioned in Sec. 4, we can consider the three types of the pooling function regarding $\lambda$ in (7) ; we apply *constant* $\lambda = 1$ as the simplest way, directly train the *parametric* $\lambda$ as in the trainable pooling [30], and estimate the *variable* $\lambda$ from the global features by the proposed GFGP (11). As shown in Table 1a, the methods that learn $\lambda$ as *parameter* and *variable* work well to improve performance, and particularly the GFGP leveraging the global information to *variable* $\lambda$ via (11) outperforms the others by a large margin. Then, the importance of the *global* information in GFGP is verified through the comparison with the method that applies (11) *locally*; that is, in (11), t is set to a *local* average pooling on the $2\times2$ receptive field to produce the variable $\lambda_p^c$ for each output $Y_p^c$. Note that this *local* method also employs the same MLP model composed of the parameters $\boldsymbol{U}$ and $\boldsymbol{V}_\lambda$ of the same size as in the GFGP (11). The performance comparison in Table 1b demonstrates that the global method (GFGP) is superior to the local one, validating the effectiveness of the global information in the pooling method.

While the position prior was turned off ($\boldsymbol{\rho} = \mathbf{0}$) in the above analysis, we here apply the full GFGP estimating two variables of $\boldsymbol{\lambda}$ and the position priors $\boldsymbol{\rho}$; in this case of $2\times2$ pool size, the position priors are described by *four*-dimensional variable per channel resulting in $\boldsymbol{\rho} \in \mathbb{R}^{4C}$. Table 1c shows that the full method is comparable to the simple method of $\boldsymbol{\rho} = \mathbf{0}$. The sensibility to the local position via the priors $\boldsymbol{\rho}$ might prevent the pooling from increasing robustness against translation. Similar results are found in the Wavelet pooling [39] which applies wavelet filters re-

Table 1. Performance analysis for global feature guidance. The performance of the proposed GFGP is highlighted by bold fonts.

(a) Pooling type w.r.t. $\lambda$ in (7) under $\rho = 0$     original

| $\lambda$ | Constant | Parameter | **Variable (GFGP)** | max-pool |
|---|---|---|---|---|
| Top-1 | 29.82 | 29.56 | **28.57** | 30.02 |
| Top-5 | 10.40 | 10.26 | **9.65** | 10.27 |

| (b) Receptive field in (11) | | | (c) Position prior | | |
|---|---|---|---|---|---|
| t | **global** avg | local avg | Variable $\lambda$ | | $\lambda, \{\rho_q\}_q$ |
| Top-1 | **28.57** | 29.02 | **28.57** | | 28.62 |
| Top-5 | **9.65** | 9.92 | **9.65** | | 9.57 |

lated to the position prior (Sec. 3.3) and produces comparable performance with the simple average pooling. Thus, in this work, we apply the simple GFGP without priors $\boldsymbol{\rho}$ in (7) which contains only $C$-dimensional variable $\boldsymbol{\lambda}$ to be estimated by (11).

### 5.1.2  MLP model in GFGP

Next, we analyze the MLP model (12) to map the global features into the pooling parameters $\{\lambda^c\}_{c=1}^C$.

**Global statistics t.** There are three standard models of global *average (avg)*, *standard deviation (std)* and *max* for the statistics t, which are compared in Table 2a; note that the statistics of *avg+std* produce doubled feature dimensionality ($k = 2$ in Fig. 1) compared to the others ($k = 1$). The simple global *avg* exhibits favorable performance, while *max* would be less suitable to these dense feature maps [1] and be vulnerable to outliers in the less discriminative features at the shallower layers.

**MLP architecture.** Table 2b compares the MLPs of various numbers of hidden nodes $D$ as well as the single layered perceptron (SLP) of $\mathsf{s}(\boldsymbol{U}\mathsf{t}(\boldsymbol{X}))$. Under the same number of total parameters, $C^2$, the MLP-$\frac{1}{2}$ outperforms the SLP due to the non-linearity of the MLP mapping. Though MLP-1 doubling the number of hidden nodes $D$ slightly improves the performance, the MLP-$\frac{1}{2}$ is favorable based on the trade-off between performance and computation cost.

**Activation function s.** The range of $\lambda$ is restricted via the activation function s, for which we can consider four types of functions; sigmoid $\frac{1}{1+\exp(-x)} \in [0,1]$, softplus $\log(1 + \exp(x)) \in [0, +\infty]$, hyperbolic tangent (tanh) $\frac{\exp(x)-\exp(-x)}{\exp(x)+\exp(-x)} \in [-1,1]$, and identity $x \in [-\infty, +\infty]$. Note that the latter two functions might push the pooling toward *min*-pooling via $\lambda < 0$. The scale of $\lambda$ is bounded by tanh and sigmoid functions, letting the scale factor in (7) rely on the features $\boldsymbol{X}$ which would be properly trained in an end-to-end manner; if the features are normalized, the scale factor should be embedded into $\lambda$ as in [37]. We can see in Table 2c that the sigmoid and softplus functions work well, producing non-negative $\lambda$ to push the pooling (7) toward avg/max-pooling. This result implies that the min-pooling suppressing the feature channel degrades per-

Table 2. Performance comparison across various settings in the MLP model (12).

(a) Global statistics t

| Statistics | **avg** | avg+std | max |
|---|---|---|---|
| Top-1 | **28.57** | 28.56 | 28.93 |
| Top-5 | **9.65** | 9.70 | 10.01 |

(b) Mapping architecture

| Architecture | SLP | MLP-$\frac{1}{4}$ | **MLP-$\frac{1}{2}$** | MLP-1 |
|---|---|---|---|---|
| size of $U$ | $C \times C$ | $\frac{C}{4} \times C$ | $\frac{C}{2} \times C$ | $C \times C$ |
| size of $V$ | - | $C \times \frac{C}{4}$ | $C \times \frac{C}{2}$ | $C \times C$ |
| Top-1 | 28.72 | 28.80 | **28.57** | 28.36 |
| Top-5 | 9.71 | 9.83 | **9.65** | 9.51 |

(c) Activation function s

| s | **sigmoid** | softplus | tanh | identity |
|---|---|---|---|---|
| Range of $\lambda$ | $[0, 1]$ | $[0, +\infty]$ | $[-1, 1]$ | $[-\infty, +\infty]$ |
| Top-1 | **28.57** | 28.63 | 28.96 | 28.70 |
| Top-5 | **9.65** | 9.75 | 9.75 | 9.80 |

Table 3. Performance comparison under the same number of additional parameters, $C^2$ per pooling layer.

| Model | NiN [22] | ResNiN [22] | SE [14] | **Ours** |
|---|---|---|---|---|
| Top-1 | 30.61 | 29.12 | 29.24 | **28.57** |
| Top-5 | 10.80 | 10.04 | 10.03 | **9.65** |

Table 4. Performance analysis regarding depths where the proposed pooling substitutes for the original max-pooling. '✓' indicates the replacement by the proposed one, while '-' means the original max-pooling.

| | \multicolumn{5}{c}{Pooling layer (Depth)} | | | | | \multicolumn{2}{c}{Error rate (%)} | |
|---|---|---|---|---|---|---|---|
| | 1st | 2nd | 3rd | 4th | 5th | Top-1 | Top-5 |
| (i) | ✓ | - | - | - | - | 30.01 | 10.47 |
| (ii) | ✓ | ✓ | - | - | - | 29.68 | 10.38 |
| (iii) | ✓ | ✓ | ✓ | - | - | 29.43 | 10.24 |
| (iv) | ✓ | ✓ | ✓ | ✓ | - | 29.08 | 9.89 |
| (v) | ✓ | ✓ | ✓ | ✓ | ✓ | **28.57** | **9.65** |
| (vi) | - | ✓ | ✓ | ✓ | ✓ | 28.69 | 9.61 |
| (vii) | - | - | ✓ | ✓ | ✓ | 29.04 | 9.83 |
| (viii) | - | - | - | ✓ | ✓ | 28.92 | 9.90 |
| (ix) | - | - | - | - | ✓ | 29.71 | 10.17 |

formance, while avg/max-pooling excites the feature channel to favorably improve the performance. The best performance is produced by the sigmoid which properly limits the range of $\lambda$ as well as excludes the min-pooling.

### 5.1.3 Effectiveness by increased number of parameter

Our pooling layer is efficiently computed due to the global average pooling followed by the MLP as in SE [14], computation cost of which might be further improved such as by grouped convolution [40] and channel shuffling [46].

As to the network size, the proposed pooling method introduces only $C^2$ additional parameters per pooling layer, as shown in Table 2b. From the viewpoint of the increased number of network parameters, we show the effectiveness of the proposed method in comparison with the other types of layers that adds the same number of parameters; NiN [22] based on $1 \times 1$ conv, ResNiN which adds an identity path to the NiN module as in ResNet [12], and squeeze-and-excitation (SE) [14], whose detailed structures are shown in the supplementary material. For fair comparison, these methods are implemented by using the same MLP-$\frac{1}{2}$ as ours (Table 2b) of $C^2$ parameters and are embedded so as to work on the feature map fed into the (original) *max* pooling layer; note that those comparison modules do not provide the functionality of downsizing feature map. The performance results are shown in Table 3, demonstrating that the proposed pooling method benefits from the additional parameters most effectively. And, it should be noted that for further improving performance, the proposed pooling method could work in conjunction with these modules that enhance the discriminativity of the feature map without pooling functionality.

### 5.1.4 Depth

We investigate the effect of our pooling method in terms of the depth where it is embedded. In this experiment, the method replaces some of the max-pooling layers in VGG-13, while in the other experiments the CNN is fully equipped with the proposed method at all the pooling layers. The performance results in Table 4 show that our method at the deeper layers contributes to performance improvement more effectively; removing the proposed pooling at the shallower layers (Table 4vi~viii) degrades performance only slightly. The deeper layers produce the more discriminative features on which the flexible pooling works well. Thus, it would be effective to apply the proposed pooling method only at the deeper layers, for suppressing the increase of parameters in CNNs.

### 5.2. Analysis of variable $\lambda$

We then analyze the contents of $\boldsymbol{\lambda}$ produced by (11) in the proposed GFGP. At each local pooling layer, the produced $\{\lambda^c\}_{c=1}^C$ are distributed over $[0, 1]$ due to the sigmoid activation s and we quantize the distribution into three-dimensional histogram of three bins as shown in Fig. 3. Since the sum of the histogram counts is constantly $C$, the three-dimensional histograms are lying on 2-simplex as shown in Fig. 2; we randomly picked up 10,000 samples from the ImageNet dataset and feed-forward them through the trained VGG-13 to obtain 10,000 histogram vectors at each pooling layer. At the first layer, one can see various types of the distribution of $\lambda$ which are spread diversely around the center[1]. It indicates that the optimal pooling

1st layer ($C = 64$)   2nd layer ($C = 128$)   3rd layer ($C = 256$)   4th layer ($C = 512$)   5th layer ($C = 512$)

Figure 2. Visualization of the distribution of $\{\lambda^c\}_{c=1}^{C}$ produced by (11) where $C$ is the number of channel at each pooling layer. The distribution of $C$ samples $\{\lambda^c\}_{c=1}^{C}$ is quantized into three bins to form three-dimensional histogram (Fig. 3) which lies on 2-*simplex*. Each point indicates each distribution (histogram of three bins) and is colored by assigning to RGB-channels the frequencies on the three histogram bins at the first layer. The center point ('×') means a uniform distribution. This figure is best viewed in color.

types vary from sample to sample. On the other hand, at the deeper layers, the distribution of $\lambda$ is somehow biased, and in particular, the last fifth layer exhibits high bias toward the pooling type of either average or max. The last pooling layer receives discriminative features which might require the distinctive pooling type such as average/max-pooling. It would lead to the effectiveness of the proposed pooling at the deeper layers as discussed in Sec. 5.1.4.

Then, to measure the dependency of $\lambda$ on the object categories, we show in Table 5a the Fisher discriminant score by applying the Fisher discriminant analysis to the three-dimensional histogram vectors which are provided with the class labels. All the layers exhibit low discriminant scores, indicating that the produced $\lambda$ are less dependent on object categories. Actually, in Fig. 3 which shows examples of the distribution of $\lambda$, we can see clear difference between two samples belonging even to the identical ImageNet category.

We also analyze the relationship between layers in terms of $\lambda$. Each sample image produces the distributions of $\lambda$ at respective layers, and the correlation coefficients between layers are computed by canonical-correlation analysis over the three-dimensional histogram vectors. As shown in Table 5b, although the adjacent two layers exhibit relatively high correlations, they are generally low, showing less correlation among the layers.

Thus, we can conclude that it is important to produce the variable $\lambda$ at each layer for each sample without sharing it across layers nor considering class categories.

## 5.3. Comparison to the other methods

Finally, the proposed method is compared with the other pooling methods on several deep CNNs. We first consider the CNNs of VGG-16 [32] which contains five local max-pooling layers and VGG-16-mod [17] of four local max-pooling layers and one global average-pooling; all the local *max*-pooling layers are implemented with $2 \times 2$ pool size and $(2,2)$-striding. We replace those local pooling with the other types of pooling for comparison: *skip* pooling implemented by the convolution with $(2,2)$-striding, *average*-pooling, two types of *stochastic* pooling methods [44, 45],

Figure 3. Examples of the distributions of $\{\lambda^c\}_{c=1}^{C}$. The distribution is quantized into a histogram of three bins over $[0, 1]$. These two histograms are produced by two image samples belonging to the identical ImageNet category.

Table 5. Statistics for the distribution of $\lambda$. (a) The higher score means the higher class-dependent distribution of $\lambda$. (b) The higher correlation indicates the closer connection regarding the distribution of $\lambda$ between two layers.

(a) Fisher discriminant score

| 1st layer | 2nd layer | 3rd layer | 4th layer | 5th layer |
|---|---|---|---|---|
| 0.1395 | 0.2021 | 0.1854 | 0.2276 | 0.1610 |

(b) Correlation coefficient

|  | 1st layer | 2nd layer | 3rd layer | 4th layer | 5th layer |
|---|---|---|---|---|---|
| 1st | - | 0.4400 | 0.2451 | 0.1739 | 0.1445 |
| 2nd | - | - | 0.3172 | 0.2165 | 0.1771 |
| 3rd | - | - | - | 0.3454 | 0.2581 |
| 4th | - | - | - | - | 0.3795 |

three trainable pooling methods of *DPP* [30], *Gated* pooling [21] and our pooling (7) of *parametric* $\lambda$, and the proposed GFGP which applies (11) to the pooling (7). The deep CNNs are trained on ImageNet dataset in the same way as in Sec. 5.1; the detailed procedures to embed the proposed pooling into CNNs and train them are shown in the supplementary material.

The performance results are shown in Table 6a. The *skip* pooling which depends on the local *position* is inferior to the others due to the position sensitivity discussed in Sec. 5.1.1. And, the stochastic approaches [44, 45] are less effective and especially the S3-pooling [45] which renders further randomness to the pooling is unsuitable for this task

on large-scale dataset. Such a randomness in pooling would hamper training of the networks, and the deterministic way works on the large-scale dataset which contains large variation in training images with enough data augmentation. The proposed GFGP favorably outperforms the standard pooling methods as well as the sophisticated ones [30, 21].

We then evaluate the pooling methods on the deeper CNNs of ResNet-50 [12] and ResNeXt-50 [40]. These models contain one *skip* pooling of $(2, 2)$-striding at the first convolution layer, followed by *max*-pooling with $3 \times 3$ pool size and $(2, 2)$-striding, and three *skip* pooling of $(2, 2)$-striding in the three ResBlocks, respectively; there are totally five local pooling layers to be replaced with the other pooling methods. In these deeper CNNs, the (original) skip-pooling is again inferior even to the simple avg/max-pooling as in Table 6a. The performance is considerably improved by the proposed GFGP with (7); on ResNet, it is superior even to the further deeper CNN of ResNet-101 which produces 22.48% (top-1) and 6.43% (top-5).

In addition to the above performance comparison, we also extends the previous trainable pooling methods [30, 21] by applying the global feature guidance (Sec. 4) to estimate the trainable parameters via the mapping (11); the DPP [30] contains two parameters per channel while the Gated pooling [21] has one parameter per channel. Our extension[2] favorably boosts the performance as shown in Table 6, demonstrating the generality of the GFGP approach (Fig. 1) .

The proposed method is also evaluated on the Places-365 dataset [47] for scene classification, the different task from the object recognition in the ImageNet. We apply the same CNN models as in Table 6 by replacing all the local pooling layers with our GFGP as well. The performance results in Table 7 demonstrate the effectiveness of the proposed method on the task of scene classification, which shows the applicability of the proposed pooling to versatile tasks.

As shown in the above experimental results, the proposed method generally boosts the performance of deep CNNs by simply replacing the local pooling layers. Since the proposed method operates only on the pooling layers, it is noteworthy that the method could favorably work with the techniques applied to refine the feature map, as discussed in Sec. 5.1.3.

## 6. Conclusion

In this paper, we have proposed a flexible pooling method adaptively tuned based on input features. The proposed method is composed of both a parameterized pooling function derived from the probabilistic perspective of maximum entropy principle and an adaptive way to estimate the

Table 6. Performance comparison on ImageNet dataset [6]. The pooling method marked by $^*$ is the original setting in the CNN model; skip$^*$ in (b) indicates the original setting without manipulating any pooling layers in the models.

(a) VGG-based models

| Pooling | VGG-16 [32] top-1 | top-5 | VGG-16-mod [17] top-1 | top-5 |
|---|---|---|---|---|
| skip | 29.60 | 10.16 | 26.00 | 8.26 |
| avg | 28.44 | 9.53 | 25.50 | 8.01 |
| max$^*$ | 27.94 | 9.25 | 25.66 | 7.97 |
| stochastic [44] | 28.66 | 9.67 | 25.74 | 8.18 |
| S3 [45] | 35.38 | 13.76 | 29.45 | 10.46 |
| DPP [30] | 28.39 | 9.45 | 25.55 | 8.04 |
| Gate [21] | 28.06 | 9.38 | 25.20 | 8.01 |
| (7) of parametric $\lambda$ | 27.92 | 9.19 | 25.42 | 7.94 |
| GFGP with (7) | 27.17 | 8.77 | 24.63 | 7.50 |
| GFGP with DPP [30] | 28.03 | 9.23 | 25.08 | 7.81 |
| GFGP with Gate [21] | 27.36 | 9.00 | 24.82 | 7.48 |

*(left-margin grouping labels: simple / stochastic / trainable / ours)*

(b) ResNet-based models

| Pooling | ResNet [12] top-1 | top-5 | ResNeXt [40] top-1 | top-5 |
|---|---|---|---|---|
| skip$^*$ | 23.53 | 7.00 | 22.69 | 6.65 |
| avg | 22.61 | 6.52 | 22.14 | 6.35 |
| max | 22.99 | 6.71 | 22.20 | 6.24 |
| DPP [30] | 22.52 | 6.63 | 21.84 | 5.98 |
| Gate [21] | 22.27 | 6.33 | 21.63 | 5.99 |
| GFGP with (7) | 21.79 | 5.95 | 21.35 | 5.74 |
| GFGP with DPP [30] | 22.66 | 6.60 | 21.79 | 6.02 |
| GFGP with Gate [21] | 22.20 | 6.26 | 21.45 | 5.81 |

*(left-margin grouping labels: simple / trainable / ours)*

Table 7. Performance comparison on Places-365 dataset [47].

(a) VGG-based models

| Pooling | VGG-16 [32] top-1 | top-5 | VGG-16-mod [17] top-1 | top-5 |
|---|---|---|---|---|
| max$^*$ | 46.25 | 15.95 | 45.44 | 15.11 |
| GFGP with (7) | 45.99 | 15.46 | 45.33 | 14.96 |

(b) ResNet-based models

| Pooling | ResNet [12] top-1 | top-5 | ResNeXt [40] top-1 | top-5 |
|---|---|---|---|---|
| skip$^*$ | 44.88 | 14.62 | 44.52 | 14.36 |
| GFGP with (7) | 44.07 | 13.94 | 44.25 | 13.94 |

parameters by means of the global feature statistics. The parameters in the pooling function flexibly controls the pooling type toward such as average and max-pooling. And, the global information is effectively incorporated into the local pooling function through adaptively tuning the pooling type (parameters) based on the input global features. We performed thorough experiments to present an effective form of the proposed pooling method and demonstrate favorable performance on large-scale image classification tasks using ImageNet and Places-365 datasets.

## Footnotes

[1]The center point corresponds to the uniform distribution of $\{\lambda^c\}_{c=1}^C$.

[2]As suggested in the papers [30, 21], we employ a exponential activation function for s in the GFGP-DPP, while the sigmoid is applied to s in the GFGP-Gated. The other settings in (11) are the same as our GFGP.

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

[Supplementary Material 2]

# Supplementary material for
# Global Feature Guided Local Pooling

This material details the implementation of the models used in the submitted paper.

## A. Embedding the proposed pooling into CNNs

We can straightforwardly replace the pooling layers *explicitly* embedded in CNNs, such as avg/max-pooling, with the proposed pooling layer of the same pool size and the striding step size; see Fig. A*ii*&*iii* in which *skip*-pooling is replaced with the proposed one. Those explicit pooling layers are found in such as VGG models [8] and the second layer (of max-pooling) in ResNet models [3, 10]. On the other hand, in the ResNet models, the *strided* convolution *implicitly* contains a pooling to downsize the feature map, and the proposed pooling is also applicable to it as follows.

### A.1. Strided convolution

The *strided* convolution (Fig. A*i*) is decomposed into the ordinary convolution without striding and the *skip*-pooling as shown in Fig. A*ii*. The *skip*-pooling simply picks up one neuron activation in the receptive field and pass it to the next layer; for example, $(2, 2)$-striding in convolution corresponds to the *skip*-pooling of $2 \times 2$ pool size with $(2, 2)$ striding step size, which picks up the top-left corner neuron activation in the $2 \times 2$ local region. The *skip*-pooling is then subject to the replacement by the proposed pooling (Fig. A*iii*).

### A.2. ResBlock

The ResNet models [3, 10] stack the residual blocks, denoted by ResBlocks, containing several convolution layers and a shortcut path. For downsizing a feature map, the ResBlock is implemented as shown in Fig. B*i* which leverages *strided* convolution operations in the both paths. By decomposing the *strided* convolutions, without changing the output, the ResBlock results in the form that explicitly applies the *skip*-pooling (Fig. B*ii*). Then, the proposed pooling substitutes for the *skip*-pooling lying at the last layer of the ResBlock as depicted in Fig. B*iii*.

## B. Embedding the comparison modules into VGG

In Sec. 5.1.3, the proposed method is compared to the modules of NiN [7], ResNiN [7, 3] and squeeze-and-excitation (SE) [4]. Those modules are embedded into the VGG model as shown in Fig. C. It should be noted that, for fair comparison, (1) they are inserted just before the max-pooling and (2) they contain the same number of additional parameters by using the same structure of MLP as ours (Fig. A*iii*); the NiN and ResNiN applies the ReLU at the last activation in stead of sigmoid.

## C. Training procedure

All the CNN models are implemented by using MatConvNet [9] and for fair comparison they are trained from scratch on NVIDIA Tesla P40.

The VGG-based models [8, 6] are trained by following the training procedure presented in MatConvNet toolbox which provides a good practice to train those models of high performance while requiring smaller number of training epochs; SGD is applied to train the CNNs with the batch size of 64, the momentum of 0.9, the weight decay of 0.0005, and the learning rate reducing from 0.1 to 0.0001 linearly in a log-scale over 20 training epochs.

The ResNet-based models [3, 10] are also trained based on the approach presented in [3]; we apply the batch size of 256 and SGD with momentum of 0.9 and weight decay of 0.0001, while the learning rate starts from 0.1 and is divided by 10 every 30 epochs throughout 100 training epochs.

Practically, we can obtain the performance close to the reported ones [1, 2]; for VGG-16, ours are 27.94% (top-1) and 9.25% (top-5) while the reported ones are 28.3% (top-1) and 9.5% (top-5); for ResNet-50, ours are 23.53% (top-1) and 7.00% (top-5) while the reported ones are 24.6% and 7.7% (top-5); for ResNeXt-50, ours are 22.69% (top-1) and 6.65% (top-5) while the reported ones are 22.60% (top-1) and 6.49% (top-5).

Figure A. *Strided* convolution of $(2, 2)$ step size is replaced by the proposed pooling method. '/2' indicates the striding step size of $(2, 2)$ in the convolution or pooling operation in which the filter or pooling size is shown in the form of '$* \times *$'. The size of feature map passed through layers is also denoted by '$Height \times Width \times Channel$'. The strided convolution (*i*) corresponds to the form (*ii*) explicitly using the skip pooling. The parametric pooling in (*iii*) is implemented such as by Eq.(7) (in the submitted paper) with $\rho = 0$. Note that the Batch-Normalization [5] is applied right after each convolution layer.

Figure B. ResBlock that downsizes an input feature map is modified by embedding the proposed pooling. In (*ii*, *iii*), the gray-colored box indicates the modified or added layer, compared to the original architecture (*i*).

Figure C. Comparison modules (Sec. 5.1.3) which contain the same number of additional parameters as ours (Fig. A*iii*). Those modules are inserted just before the max-pooling since they do not render pooling functionality.