[Reviews · NeurIPS 2019]

Reviewer 1



Originality. This paper viewed the existing pooling method in a convex combination of local feature activations. Based on this model, the authors explained how the proposed Gaussian pooling is different from existing pooling methods clearly. Also, this paper proposes to modify the Gaussian distribution such that the pooling value becomes larger than mean based on the knowledge of local pooling. To the best of my knowledge, such pooling functions are novel. Quality. The quality of this paper is good. The proposed algorithm is reasonable and technically sound. The experiments are conducted on large-scale datasets and compared with related pooling methods. Minor problem: The results of stochastic [32] are missing in Table 3.(c) and (d). Clarity. This paper is clearly written except for some points described below. More explanation of the inverse softplus function and iSP-Gaussian distribution would improve clarity. Mainly, why the term exp(x)/(exp(x) – 1) exists in Eq.(10) is not clear enough. It would be better to explain how to derive Eq.(10). In the experimental section, which layers the authors applied to the proposed methods are not clear. According to the discussion of Global pooling (line 219-), it seems that the poo1 and/or pool2 of Table (a) are used in previous comparisons, but there are no explanations. The fact that this paper is only focusing on local pooling is not clearly explained until this section. Significance. The new pooling method is useful for improving the recognition accuracy of various recognition problems. This paper proposes a novel form of pooling by modifying the parameter of Gaussian distribution, which is shown effective than state-of-the-art poolings on the large-scale dataset. Thus, other researches or practitioner can use the proposed method for any algorithms based on CNN.

Reviewer 2



Updates: I appreciate the additional experiments and clarifications in the rebuttal. I think this is a good paper and would like to increase the rating. Overall, the paper is clearly written and easy to follow. It proposes an interesting novel approach to pooling that leads favorable gains in performance. Although the core mechanism, i.e. estimating pooling parameters using global features, is from the previous GFGP method [1], I think connecting it to probabilistic models is not trivial and can be regarded as a satisfactory technical contribution. My biggest concern is about its practical usefulness. It says that the proposed pooling method requires additional O(C^2) parameters, which are not negligible. For example, we could simply use more convolution filters with average/max pooling to improve performance. It would be more convincing if authors can somehow compare methods with roughly the same number of parameters to prove that the improvement is not just about the increase of model capacity in general by adding more parameters. I'd like to know more details of the derivation of the approximation in Eq.15, particularly how these fixed number are derived. Also, Eq.16 is confusing because this only applies to the case sigma_0 = 0 but not in general. Authors mention the log-Gaussian as a possible alternative to iSP-Gaussian, but no reported in experiments. Was it totally impossible to train log-Gaussian based model because of instability?

Reviewer 3



Originality: The proposed pooling function is novel. The authors show the result of their pooling function with several network architectures on several datasets including ImageNet dataset. Quality: The submission is technically sound. The authors showed that their pooling function outperforms existing pooling functions like max or average pooling. The section 3.3 shows how change the estimated parameters \mu_0 and \sigma_0 of the pooling function during training. Clarity: The paper is well written and well organized. Significance: The experiments show that the proposed pooling function is better than standard pooling functions like max and average pooling. The improvement is about 1% on ImageNet dataset. --------------------------------------- The rebuttal addresses my concerns and I think it is a good paper.

[Author Response · NeurIPS 2019]

We thank the reviewers for their valuable comments and suggestions, and list our responses as follows.

**To Reviewer #1:**

**1.** The performance results of the stochastic pooling [32] are shown in Table A and will be included in the revised paper.

**2.** Eq. 10 is derived through the following variable transformation. Suppose $y$ is a random variable whose probability
density function is Gaussian, $\mathtt{q}(y) = \frac{1}{\sqrt{2\pi}\sigma_0} \exp\{-\frac{1}{2\sigma_0^2}(y-\mu_0)^2\}$. The target random variable $x$ is obtained via softplus
transformation by $x = \mathtt{softplus}(y) \Leftrightarrow y = \mathtt{softplus}^{-1}(x) = \log[\exp(x)-1]$. Then, we apply the relationship of
$\mathtt{q}(y)dy = \mathtt{p}(x)dx$ and $\frac{dy}{dx} = \frac{\exp(x)}{\exp(x)-1}$ to provide $\mathtt{p}(x) = \frac{1}{\sqrt{2\pi}\sigma_0}\frac{\exp(x)}{\exp(x)-1}\exp\{-\frac{1}{2\sigma_0^2}(\log[\exp(x)-1]-\mu_0)^2\}$ (Eq.10).
Such detailed explanation about Eq.10 will be added to the revised paper.

**3.** We will clearly describe that this work focuses on local pooling and the method is applied to all the local pooling
layers in a CNN; e.g., pool1&2 in Table 2a of 13-layer Net.

**To Reviewer #2:**

**1.** From the viewpoint of the increased number of parameters, we show the effectiveness of the proposed method in
comparison with the other types of modules that adds the same number of parameters; NiN [LCY14] using $1 \times 1$
conv, ResNiN which adds an identity path to the NiN module as in ResNet [7], and squeeze-and-excitation (SE)
module [HSS18]. For fair comparison, they are implemented by using the same 2-layer MLP as ours (Eq.12) of $C^2$
parameters with appropriate activation functions and are embedded before pool1&2 layers in the 13-layer Net (Table
2a) so as to work on the feature map fed into the *max* pooling layer; the detailed architecture is shown in the left-bottom
figure. The performance results are shown in Table B, demonstrating that our method most effectively leverages the
additional parameters to improve performance. This comparison result will be included in the revised paper.

**2.** The approximation in Eq.15 is *heuristically* determined so as to represent $\mathrm{E}[\eta]$ in a simple analytic form. That is, under
the condition of $\sigma_0 \leq 1$, we *manually* tune the form and the parameters of the residual term, $0.115\sigma_0^2\frac{4\exp(0.9\mu_0)}{(1+\exp(0.9\mu_0))^2}$,
toward minimizing the residual error between $\mathtt{softplus}(\mu_0)$ and $\int\log[1+\exp(\tilde{\epsilon})]\mathcal{N}(\tilde{\epsilon};\mu_0,\sigma_0)d\tilde{\epsilon}$ which is empirically
computed by means of sampling. Then, Eq.16 is presented as the *most roughly* approximated form for Eq.15 by
*ignoring* the above-mentioned residual term which exhibits at most 0.115 residual error. The rough approximation is
introduced since it is practically useful for fast computation at inference without degrading performance (lines146-150).

**3.** In the preliminary experiment, we confirmed that the log-Gaussian makes it almost impossible to train CNNs; due to
introducing the log-Gaussian module, the training loss is not favorably reduced during the end-to-end learning.

**To Reviewer #3:**

**1.** As mentioned in lines 174-178, the computation overhead of the proposed method is caused by the GAP+MLP to
estimate the two parameters of $\{\mu_0, \sigma_0\}$ at training and only one $\mu_0$ at inference; $O(HWC)$ in GAP and $O(C^2)$ in MLP.
For example, in ResNet-50 which requires 3.86GFLOPs, our method increases the computation by only 0.017GFLOPs.

**2.** Table C shows the performance of ResNet-50 on the adversarial attack via FGSM [GSS15] which adds perturbation
by $\epsilon\,\mathtt{sign}(\nabla_I\mathcal{L}(I,t))$ to an input (test) image $I$ according to its label $t$ and the loss function $\mathcal{L}$. Compared to the other
pooling methods, our method exhibits favorable robustness against the attack while the Mixed pooling endowed with
stochastic training also works well. This result motivates our future work to further analyze the proposed pooling
method, especially in terms of stochastic training in the pooling, from this viewpoint of robustness to input perturbations.

Table A: Performance on ImageNet.

| Method | ResNet-50 Top-1 | ResNet-50 Top-5 | ResNeXt-50 Top-1 | ResNeXt-50 Top-5 |
|---|---|---|---|---|
| Stochastic | 25.47 | 7.87 | 25.02 | 7.73 |
| iSP-Gauss | 21.37 | 5.68 | 20.66 | 5.60 |

Table B: Performance comparison on Cifar100 dataset by 13-layer Net.

| Method | Error (%) |
|---|---|
| NiN | 24.49±0.13 |
| ResNiN | 24.33±0.16 |
| SE | 23.99±0.07 |
| iSP-Gaussian | 23.52±0.37 |

← Module architecture of the comparison methods. They utilize the same 2-layer MLP as in our method.

Table C: Performance results of ResNet-50 on ImageNet dataset through adversarial attack by FGSM. $\epsilon = 0$ means *no* adversarial attack, producing to the original results in Table 3c.

| Method | $\epsilon = 0$ Top-1 | $\epsilon = 0$ Top-5 | $\epsilon = 0.1$ Top-1 | $\epsilon = 0.1$ Top-5 | $\epsilon = 0.2$ Top-1 | $\epsilon = 0.2$ Top-5 | $\epsilon = 0.3$ Top-1 | $\epsilon = 0.3$ Top-5 |
|---|---|---|---|---|---|---|---|---|
| skip | 23.53 | 7.00 | 42.89 | 14.64 | 56.58 | 22.20 | 66.25 | 28.73 |
| avg | 22.61 | 6.52 | 40.35 | 13.22 | 53.99 | 20.13 | 63.97 | 26.62 |
| max | 22.99 | 6.71 | 45.39 | 15.41 | 60.93 | 23.59 | 71.03 | 30.64 |
| Mixed | 23.32 | 6.77 | 37.55 | 12.11 | 49.83 | 17.90 | 58.99 | 23.27 |
| DPP | 22.52 | 6.63 | 42.70 | 14.02 | 58.12 | 21.88 | 68.77 | 28.79 |
| Gated | 22.27 | 6.33 | 41.23 | 13.29 | 55.84 | 20.66 | 66.41 | 27.58 |
| GFGP | 21.79 | 5.95 | 38.11 | 11.85 | 50.44 | 17.70 | 60.06 | 23.26 |
| iSP-Gaussian | 21.37 | 5.68 | 37.42 | 11.27 | 50.26 | 17.52 | 60.02 | 23.24 |

**References**
[LCY14] M. Lin, Q. Chen, and S. Yan. Network in Network. In ICLR, 2014.
[HSS18] J. Hu, L. Shen, and G. Sun. Squeeze-and-Excitation Networks. In CVPR, pp. 7132-7141, 2018.
[GSS15] I. Goodfellow, J. Shlens, and C. Szegedy. Explaining and Harnessing Adversarial Examples. In ICLR, 2015.


[Meta-Review · NeurIPS 2019]

Reviewers found this paper well explained and to contain interesting ideas for pooling layers in CNNs. The method was applied to realistic large-scale datasets. The consensus is the proposed method adds an interesting tool to the CNN training kit.